# Intrinsic surface *p*-wave superconductivity in layered AuSn₄

Wenliang Zhu [1,15], Rui Song[2,15], Jierui Huang[3,15], Qi-Wei Wang[4,15], Yuan Cao[1,15], Runqing Zhai[1], Qi Bian[5], Zhibin Shao[1], Hongmei Jing[1], Lujun Zhu[1], Yuefei Hou[6], Yu-Hang Gao[4], Shaojian Li[5], Fawei Zheng[6], Ping Zhang[6,7] ✉, Mojun Pan[3], Junde Liu [3], Gexing Qu[3], Yadong Gu[3], Hao Zhang[1], Qinxin Dong[3], Yifei Huang[3], Xiaoxia Yuan[8], Junbao He[9], Gang Li [3,10,11], Tian Qian [3,10,11] ✉, Genfu Chen [3,10,11] ✉, Shao-Chun Li [4] ✉, Minghu Pan [1,5] ✉ & Qi-Kun Xue[12,13,14] ✉

The search for topological superconductivity (TSC) is currently an exciting pursuit, since non-trivial topological superconducting phases could host exotic Majorana modes. However, the difficulty in fabricating proximity-induced TSC heterostructures, the sensitivity to disorder and stringent topological restrictions of intrinsic TSC place serious limitations and formidable challenges on the materials and related applications. Here, we report a new type of intrinsic TSC, namely intrinsic surface topological superconductivity (IS-TSC) and demonstrate it in layered AuSn₄ with $T_c$ of 2.4 K. Different in-plane and out-of-plane upper critical fields reflect a two-dimensional (2D) character of superconductivity. The two-fold symmetric angular dependences of both magneto-transport and the zero-bias conductance peak (ZBCP) in point-contact spectroscopy (PCS) in the superconducting regime indicate an unconventional pairing symmetry of AuSn₄. The superconducting gap and surface multi-bands with Rashba splitting at the Fermi level ($E_F$), in conjunction with first-principle calculations, strongly suggest that 2D unconventional SC in AuSn₄ originates from the mixture of *p*-wave surface and *s*-wave bulk contributions, which leads to a two-fold symmetric superconductivity. Our results provide an exciting paradigm to realize TSC *via* Rashba effect on surface superconducting bands in layered materials.

Topological superconductors have attracted intense attention due to the potential for Majorana-based qubits of fault-tolerant quantum computation[1–3]. One way of searching TSC, is to realize *s*-wave superconductivity on spin helical states[4], such as semiconductor nanowires with strong spin–orbit coupling (SOC) or ferromagnetic atomic chains proximitized with an *s*-wave superconductor[1,3,5–9], vortex cores in a proximitized topological insulator[4,10]. An alternative route to TSC is to realize a *p*-wave superconductor, which is an intrinsic topological superconductor; prominent candidates are Sr₂RuO₄[11–13], UPt₃[14], CuₓBi₂Se₃[15,16], iron-based superconductors[17–20] and recently discovered Kagome superconductors *A*V₃Sb₅ (where *A* is K, Rb or Cs)[21]. In these cases, TSC is evidenced by the Majorana zero-energy modes (MZM), which were spectroscopically identified as zero-energy conductance signals, localized at the ends of the one-dimensional (1D) chain or in the vortex core, or at boundaries and defects; Another evidence is spin-rotational symmetry breaking in

A full list of affiliations appears at the end of the paper. ✉e-mail: zhang_ping@iapcm.ac.cn; tqian@iphy.ac.cn; gfchen@iphy.ac.cn; scli@nju.edu.cn; minghupan@snnu.edu.cn; qkxue@mail.tsinghua.edu.cn

the SC state observed by nuclear magnetic resonance (NMR)[15,22,23] and point-contact spectra in Cu$_x$Bi$_2$Se$_3$[16]. However, the proximity-induced TSC heterostructures require a long superconducting coherence length, in principle prohibiting the use of high-temperature superconductors. Additionally, the complex heterostructures make further exploration and applications challenging. Meanwhile, the existing intrinsic TSC needs special symmetry-protected non-trivial topological bands, which place limitations on materials and are also very sensitive to disorder.

In this work, we propose a new type of intrinsic TSC, namely intrinsic surface topological superconductor (IS-TSC) as well as demonstrated it in layered material AuSn$_4$. Instead of requiring the specific topological bands, an IS-TSC is introduced by multiple surface bands with Rashba splitting across the $E_F$ and further condensing into a coherent paired state. We employ ultralow temperature scanning tunneling microscopy/spectroscopy (STM/STS), angle-resolved photoemission spectroscopy (ARPES), transport, point-contact spectroscopy (PCS), and density functional theory (DFT) calculations, to investigate intensively the atomic structure, super-conductivity, band structure and surface states of AuSn$_4$. Transport measurements reveal that the AuSn$_4$ is a superconductor with $T_c \sim 2.4$ K and the upper critical field ($H_{c2}(0)$) ~643 Oe and 1621 Oe for $H \parallel a$ and $H \parallel ab$, respectively. Two-fold symmetric super-conductivity is observed in both magneto-transport and point-contact spectroscopy with the fields rotating in $ab$ plane. A super-conducting gap develops below $T_c$ in tunneling spectrum; its temperature and field dependencies agree well with the results of transport measurements. ARPES and first-principle calculations, further visualize multiple surface bands crossing the $E_F$ as well as their Rashba splitting, which suggests the observed unconventional SC can ascribe to a mixing of $s + p$ wave pairing.

## Results

### Structure, superconductivity, and anisotropic upper critical fields of AuSn$_4$

Noble metal alloys $A$Sn$_4$ ($A$ = Au, Pt, and Pd), have been regarded as promising candidates of TSC owing to both topological band structure[24–27] and superconductivity[28–30]. For example, the work[31] reported the superconductivity ($T_c$ ~2.7 K) and possible non-trivial band topology in PtPb$_4$. Recent works found the superconductivity of AuSn$_4$[32] and claimed it as a 2.6 K topological superconductor with non-trivial surface states[33]. Here we grew high-quality single crystals of AuSn$_4$ by the flux method (see Methods). An optical image in Fig. 1a shows a square-shaped AuSn$_4$ crystal with ~1 mm size and ~0.1 mm thick and a large shiny surface indicating the $ab$ plane. The structure of AuSn$_4$ samples is determined by atomic-level scanning transmission electron microscopy imaging, electron diffraction image, and x-ray single crystal diffraction, as shown in Fig. 1b, c, Supplementary Tab. S1 and Figs. S1 and S2. The AuSn$_4$ has stacked Sn-Au-Sn trilayers, each Au layer is sandwiched by two Sn layers forming a square lattice (see the structural model in Supplementary Fig. S2a, b). The structure is in the space group $Aba2$ (No. 41) with the lattice constants of $a$, $b = 6.476$ Å, and $c = 11.666$ Å, distinct from the previously reported structures of PtSn$_4$ and PdSn$_4$, which hold orthorhombic ($Ccca$, No. 68) space group[34,35]. Further, the electron diffraction image (Fig. 1c) meets well with the simulated one based on the structure of space group $Aba2$ (No. 41), instead of $Ccca$ (No. 68) (see Supplementary Fig. S2c, d).

Figure 1d displays the temperature dependence of resistivity $\rho_{xx}(T)$ for AuSn$_4$ single crystals with the current applied in the $ab$ plane by the standard four-probe method at zero magnetic field. The normal state resistivity shows a metallic behavior with a moderately small residual resistivity of ~0.75 μΩ cm, and the residual resistivity ratio [RRR = $\rho_{xx}(300\,K)/\rho_{xx}(3\,K)$] is about 85, indicating high crystalline

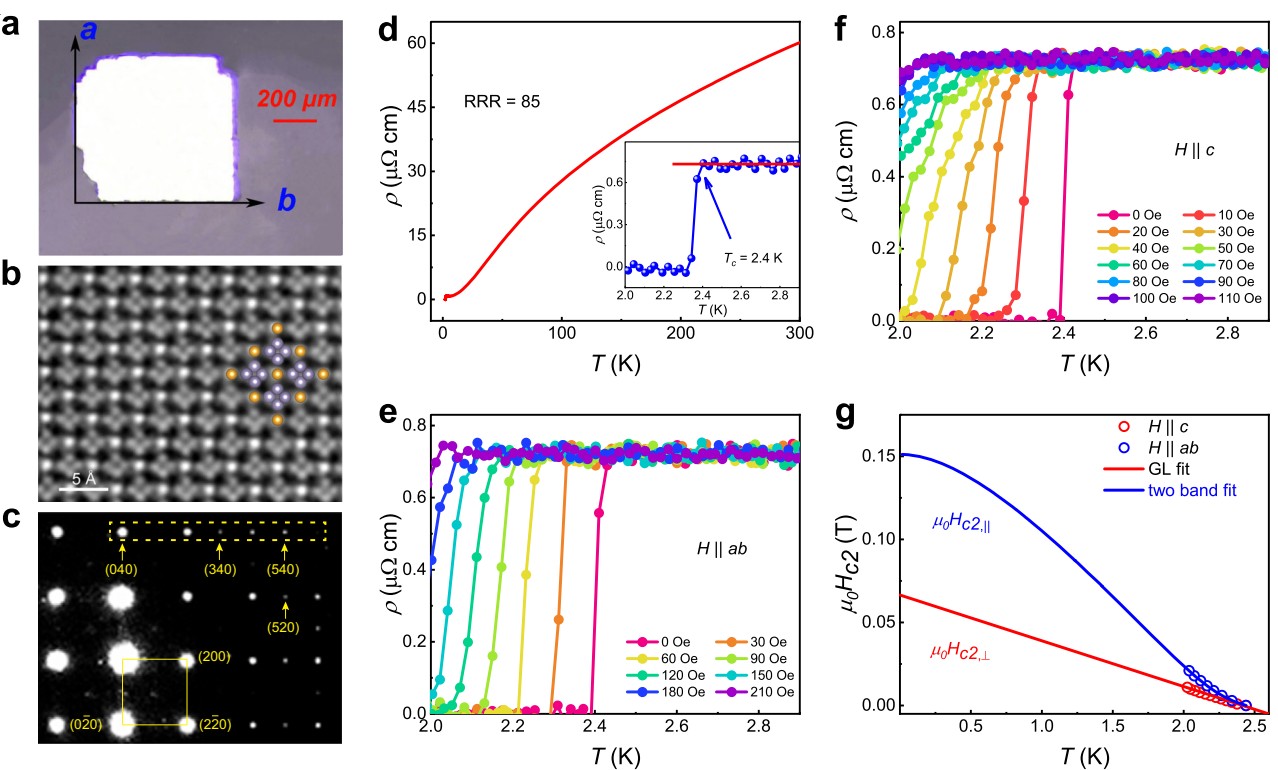

**Fig. 1 | Superconductivity and anisotropic upper critical fields of single crystal AuSn$_4$. a** An optical image of the grown AuSn$_4$ crystal. **b** High-resolution TEM image of the $ab$ plane. (inset) top view shows the square-net lattice of Sn-Au-Sn trilayer. **c** The measured electron diffraction image of AuSn$_4$ sample along the $c$ axis. **d** The temperature dependence of the resistivity with current flowing in the $ab$ plane. The inset shows $\rho$(T) in zero field at temperatures near $T_c$. **e–f** The low-temperature resistivity curves under magnetic fields with $H \parallel ab$ and $H \parallel c$, respectively. **g** Temperature dependence of the upper critical field $\mu_0 H_{c2}$ with magnetic field directions parallel ($\mu_0 H_{c2,\parallel}$) and perpendicular ($\mu_0 H_{c2,\perp}$) to the crystal plane.

quality and low density of defects. A sharp SC transition appears at the onset temperature $T_c$ ~ 2.4 K, defined by the intersection of the tangent line and the $R-T$ curves, indicated by the thin arrow (the inset of Fig. 1d). The low-temperature $R(T)$ under various magnetic fields with $H \parallel ab$ (Fig. 1e) and $H \parallel c$ (Fig. 1f), display the SC transition shifts gradually towards lower temperatures with increasing fields. Extracted from Fig. 1e, f, the temperature dependence of upper critical fields ($\mu_0 H_{c2,\parallel}$ and $\mu_0 H_{c2,\perp}$), have an upward curved feature for $H \parallel ab$ and a linear behavior for $H \parallel c$, respectively (Fig. 1g). The linear temperature dependence can be described by Ginzburg–Landau (GL) theory $H_{c2,\perp}(T) = (\phi_0/2\pi\xi_{GL}^2)(1 - T/T_c)^2$, where $\phi_0$ is the magnetic flux quantum and $\xi_{GL}$ is the zero temperature GL coherence length. As shown in Fig. 1g, the red line gives $\xi_{GL} = 71.69$ nm, roughly equal to the reported value[30], and the upper critical field at $T = 0$ K for $H \parallel c$ is accordingly estimated as $\mu_0 H_{c2,\perp}(0) = 643$ Oe. For $H \parallel ab$, $\mu_0 H_{c2,\parallel}$ deviates from the GL equation. The enhancement of $\mu_0 H_{c2,\parallel}$ at low temperatures can be interpreted by impurity-induced disorder[36], dimensional crossover[37], quantum melting of the vortex lattice[38], or multiband effect[39]. The first three are obviously irrelevant here because of high crystal quality, $\xi_{GL}$ is much larger than the layer distance and no trace for the quantum critical point. Moreover, the calculated band structure later shows that several surface bands cross the Fermi level. Therefore, $\mu_0 H_{c2,\parallel}$ is fitted by the equation for a two-band superconductor (blue curve)[40,41] with $\mu_0 H_{c2,\parallel}(0) = 1621$ Oe (see the details of two-band fitting in Supplementary Note 4), which indicates the superconductivity was more susceptible to perpendicular magnetic fields than to in-plane magnetic fields. The ratio of $\mu_0 H_{c2,\parallel}(0)/\mu_0 H_{c2,\perp}(0)$ is 2.5, suggesting a strong anisotropy and 2D nature for superconductivity.

## Two-fold symmetric magnetoresistance and point-contact spectra in superconducting regime

With standard four-probe method (Fig. 2a), we performed the angle-dependent magnetoresistance measurements by rotating magnetic field $H$ in the ab plane under the current $I$ along $a$ and $b$ axis (see the angular relationship in Supplementary Fig. S17). Figure 2b–e present the angular dependences of normalized resistance $R(\theta)$ at 1.8 K with various fields rotating in the ab plane. A two-fold symmetry are clearly observed when the fields are below the upper critical field $\mu_0 H_{c2,\parallel}$ at 1.8 K ( ~ 480 Oe), in contrast to isotropic magnetoresistance with the fields above the critical field, irrespective of the current $I \parallel a$ or $b$ axis. Meanwhile, two-fold symmetry of the resistance gradually decreases with increasing the field up to 450 Oe, inferring it originated from the superconductivity. Such pronounced two-fold symmetric resistivity is shown clearly in Fig. 2d with the dumbbell-shape only under the fields below $\mu_0 H_{c2,\parallel}$ at 1.8 K, and becomes isotropic in the normal state (see Supplementary Fig. S3 for other samples), which excludes the extrinsic oscillation from small misalignment of between crystalline plane and the field direction. Besides, a 90° angle between the $C_2$ axes of angle-dependent magnetoresistance for $I \parallel a$ and $I \parallel b$ on the same sample in Fig. 2e, indicates that it is intrinsic and independent of the current direction. The zero-resistance temperature also shows twofold modulation and a 90° angle between the $C_2$ axes for $I \parallel a$ and $I \parallel b$ (see Supplementary Fig. S18). These observations strongly evidence an unconventional pairing mechanism in AuSn₄ below $T_c$.

Point-contact spectroscopy (PCS), has been widely used to examine the pairing symmetry for unconventional SC. We prepared a soft-point contact (see schematic diagram in Supplementary Fig. S4) for the point contact measurement of AuSn₄. Temperature dependence of junction's resistance (Fig. 2f) presents a sharp superconducting drop at around 2.4 K. In particular, with soft-point contact technique[42], differential conductance spectra under a 120 Oe in-plane magnetic field were measured at $T = 1.8$ K by changing $\theta$ from 0 to 360°, as shown in Fig. 2g. Here, when $\theta = 0°$, the magnetic field $B$ is parallel to $b$ axis. The angle-dependent PCS at angles from 0 to 360° (Fig. 2g) presents a clear two-fold modulation with the dip positions

varying in a narrow energy range. Meanwhile, the normalized $dI/dV$ intensity at 6 mV (outside of the dip energy) presented in Fig. 2i, has less than a 0.2% variation versus the angle, indicating negligible anisotropy for normal state. On the contrary, the angular dependence of the normalized $dI/dV$ intensity at 0 mV (inside of the dip energy), displays a clear two-fold modulation (Fig. 2i). Consistent with the aforementioned two-fold symmetry observed in both magnetoresistance and zero-resistance temperature, which indictes these observations are clearly an intrinsic character rooted from the unconventional pairing symmetry of AuSn₄ and the superconducting state dominates the PCS. Figure 2h shows a typical PCS curve at $\theta = 80°$ with observing a zero-bias-conductance peak (ZBCP) and two side dips. Concerning about the observation of ZBCP, the heating effect can be excluded for the reasons that the sample is in SC state and no sharp-spiky dips appear at energies larger than gap[36,43,44]. Pronounced side dips and the peak without splitting, are also clearly against with the Blonder-Tinkham-Klapwijk (BTK) theory for conventional Andreev reflection[45].

Moreover, the minima of two dips reside at the energies of ±3 mV, in accordance with the gap energy measured by STS later, which strongly suggests that the observed ZBCP is intrinsic and related with the unconventional Andreev bound state (ABS)[46–48]. The ABS is a signature of unconventional SC, owing to the interference of the SC wave function at the surface[39]. Furthermore, the ZBCP with no more than 4% conductance increasing observed in the point-contact spectroscopy, indicates that the existence of unusual surface states, evidenced by ARPES results later, may evidences TSC[16].

## Surface topography, superconducting gap measured by STM/STS

High-resolution STM image on the surface terrace (Fig. 3a) shows the square-net lattice of topmost Sn-terminated layer and a quasi-one dimensional (Q1D) super modulation, resembling to the surface superstructure of cuprate $Bi_2Sr_2CaCu_2O_{8+d}$ (Bi2212)[49]. Zoom-in atomic resolution image (Fig. 3b) clarifies the existence of a $\sqrt{2} \times 2\sqrt{2}$ superstructure with the unit cell of 6.6 Å × 12.2 Å, may indicate the formation of charge density wave (CDW) state, which has not been reported for this material. Comparing the density of states measured by STS between the terrace and step edge (Supplementary Fig. S5), we confirm that surface states are prominent on the terrace and get greatly suppressed at the edge.

A typical normalized $dI/dV$ spectrum measured at 0.16 K (Fig. 3c) shows a superconductive gap feature, a deep V-shaped gap about $\Delta$ ~ 0.487 meV (defined as half of the gap width) with the depth of gap ~ 87%. The temperature and magnetic field dependence of $dI/dV$ spectra (Fig. 3d) show the suppression of the gap and increased zero-bias-conductance (ZBC) at higher temperatures and magnetic fields, indicating a typical characteristic of superconductor. To estimate the $T_c$ and critical field ($H_c$), the normalized ZBC are extracted from Fig. 3d and a rough linear fitting gives the deduced $T_c$ and $H_c$ values of ~2.64 K and 660 Oe, respectively (Fig. 3e), both values agree well with transport measurements (Fig. 1). The ratio $2\Delta(0)/k_B T_c$ ($k_B$ is the Boltzmann constant) is estimated to be ~4.7, which is in the range of the medium coupling Bardeen–Cooper–Schrieffer (BCS) superconductors. The upper critical fields $\mu_0 H_{c2,c}(0)$ at $T = 0$ K for $H \parallel c$ (~660 Oe) is deduced from STS, in accordance with the value of $\mu_0 H_{c2,\perp}(0)$ (643 Oe) from transport measurement. All confirms the observed gap is a superconducting gap. It is worth noting that there is still a 13% residue DOS at $E_F$, which may indicate there exists gap nodes/nodal lines or ungapped bands. Furthermore, we perform vortex measurement under magnetic fields, however, zero-bias-conductance (ZBC) maps do not show the existence of vortex, which may due to the overlaps of vortex[50] (see the details of analysis in Supplementary Note 18). Our STS measurements taken at 160 mK, show clearly the SC gaps are modulated with both $\sqrt{2} \times 2\sqrt{2}$ superstructure (CDW) and stripe pattern (Supplementary Fig. S16), suggesting that the observed CDW state is

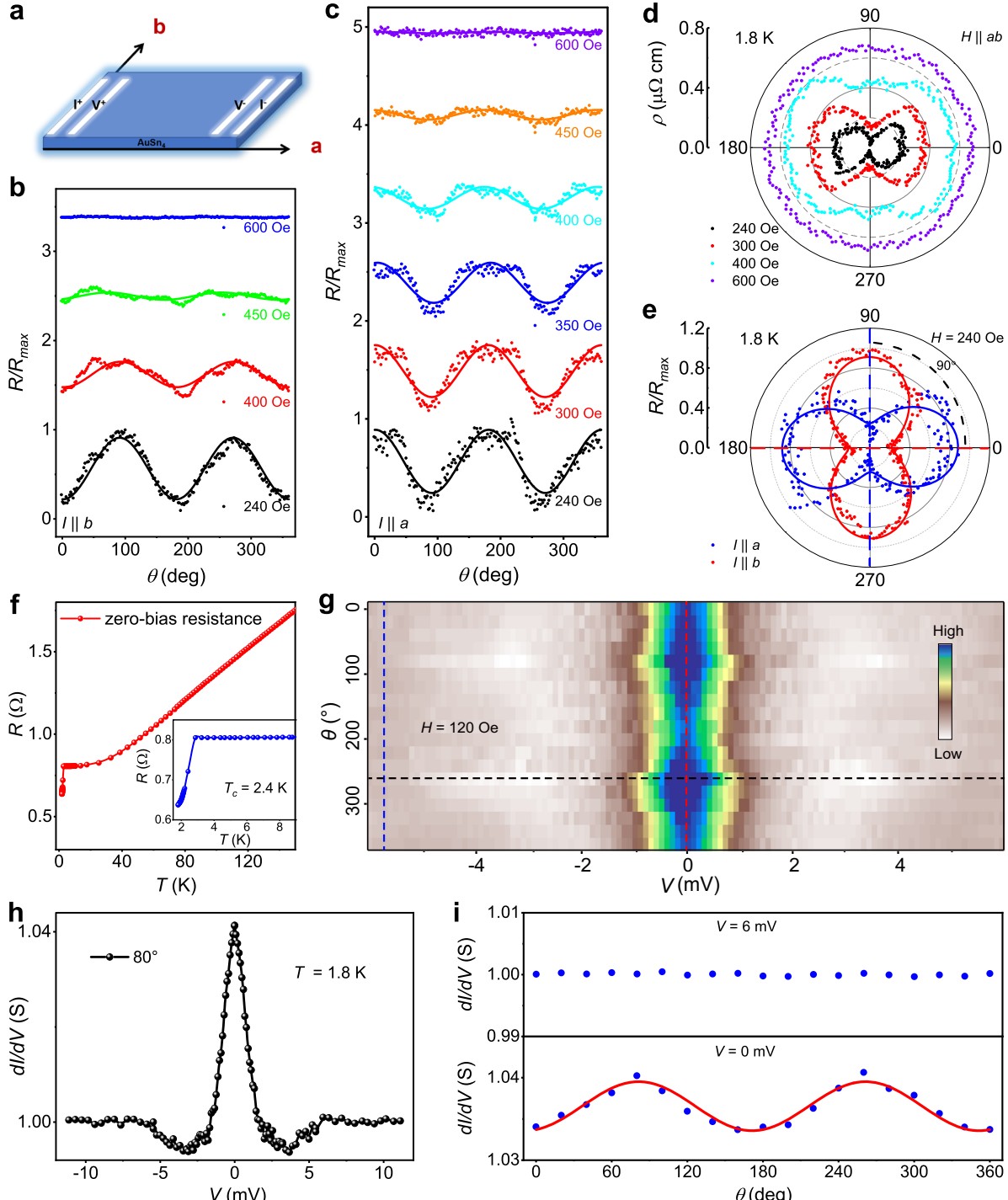

**Fig. 2 | Two-fold angular dependence of the resistance and point-contact spectra of superconducting AuSn₄ under in-plane magnetic fields. a** Schematic measurement configuration under in-plane magnetic fields. **b**, **c** The angular dependence of the normalized magnetoresistance at various in-plane magnetic fields and 1.8 K with the current along $b$ and $a$ axis, respectively. $\theta$ is the angle between the directions of fields $H$ and current $I$. **d** Polar plot of angular-dependent resistance in **c** (the radial scale is resistivity). **e** Angular-dependent normalized resistance at 240 Oe and 1.8 K with the current $I$ along $a$ and $b$ axis, respectively. Red and green dashed lines show the $C_2$ axis for $I \parallel a$ and $I \parallel b$, respectively. Solid lines in

**b**, **c**, and **e** are the fittings with a $cos(2\theta + \varphi)$ form. **f** Temperature dependence of junction's resistance for the Ag-surface of AuSn₄ sample using the modified four-point probe setup. The inset shows junctions' resistance in zero field at temperatures near $T_c$. **g** Color plot of the point-contact spectra at angles from 0 to 360° in a 120 Oe external magnetic field. **h** A representative point-contact spectra obtained on Ag-paste/AuSn₄ junction at 1.8 K (black dashed line in **g**). **i** Angle dependence of the normalized differential conductance ($dI/dV$) at selected angles from 0 to 360° at $V = 6$ and 0 mV (blue and red dashed lines in **g**), respectively. The red solid line in **i** is the best fit for a $cos(2\theta + \varphi)$ form.

relevant to the superconductivity. Interestingly, STS measurements on the flat terrace and at the edge at 160 mK (Supplementary Fig. S15), show that a slightly smaller size and shallower depth of SC gap measured at the edge, comparing to the gap measured on the flat terrace.

The depth and size of SC gap at the edge are both suppressed to 82% and 0.446 meV, compared to the values on the terrace (87.8% and 0.516 meV). The variation of SC gap due to CDW modulation on the terrace are much smaller, 0.486-0.516 meV of gap size without

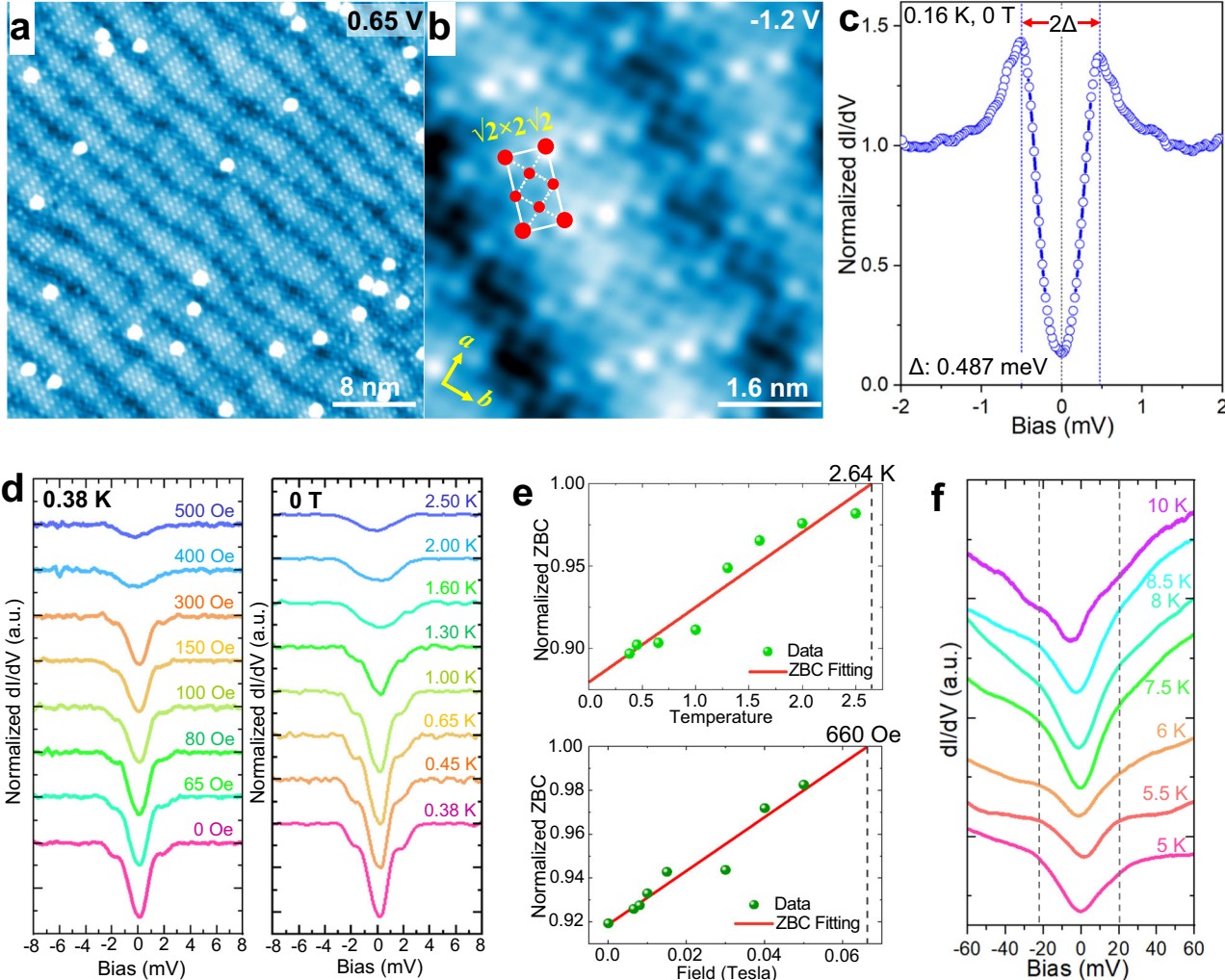

**Fig. 3 | Surface topography and the superconducting gap with variable temperatures and fields. a** High-resolution image reveals a superstructure with the modulation of wavy troughs, the image size is $40 \times 40$ nm². The bright spots on the surface can be attributed to the residual Sn from the cleavage. **b** Zoom-in atomic resolution image shows a square-net lattice and $\sqrt{2} \times 2\sqrt{2}$ superstructure with the unit cell size of (6.3-6.6) Å × (11.6-12.23) Å (6.6 Å × 13.2 Å in theory) for surface Sn lattice. **c** A representative $dI/dV$ spectrum taken at 0.16 K, showing a superconducting gap with the size and depth of gap -0.487 meV and 87%, respectively. Here, we adopt small lock-in modulation bias ($V_{\text{mod}}$) - 40 μV in order to eliminate the broadening effect. **d** A series of normalized $dI/dV$ spectra with perpendicular fields varying from 0 Oe to 500 Oe (left) and temperatures from 0.38 K to 2.5 K (right), respectively. Set points: $V_b = 10$ mV, $I_t = 400$ pA, the $V_{\text{mod}}$ is 0.3 mV. Note, the STS taken at 0.38 K with 0.3 mV $V_{\text{mod}}$, give a dual-gap feature with enlarged gap size and shallower depth, probably due to the thermal broadening effect and large lock-in modulation bias. **e** Temperature (upper) and field (lower) dependences of normalized ZBC extracted from panel **d**. The deduced $T_c$ and $H_c$ are 2.64 K and 660 Oe, respectively. **f** The gap evolution with temperatures above $T_c$.

observable change in the depth. These observations suggest the possible existence of dispersive Majorana edge mode[51]. Further elevating temperatures from 5 K to 10 K, a pseudogap-like feature develops above $T_c$ and persists up to 10 K, which may correspond to the gap of observed CDW.

## Band structure, Fermi contour, and surface states revealed by ARPES and DFT calculations

The calculated band structures of the bulk AuSn₄ is presented in Supplementary Fig. S7. Furthermore, the monolayer AuSn₄ and the surface states of bulk AuSn₄ are also calculated and shown in Fig. 4b, c, respectively, where two surface bands SS1 and SS2 near the $E_F$ are marked. Figure 4d shows the band dispersions measured along the cut of $\bar{X} - \bar{\Gamma} - \bar{X}$ direction at $k_y = 0$ (left) and the derivative ARPES plot measured along the cut of $\bar{Y} - \bar{\Gamma} - \bar{Y}$ direction at $k_x = 0$ (right), respectively, where both monolayer bands (marked by green dashed lines) and surface bands SS1 and SS2 (red dashed lines), are clearly observed and matched well with DFT calculations. Besides, the Rashba

splitting of SS1 is clearly resolved in the inset of Fig. 4d. Bulk Brillouin zone (BZ) and resultant surface BZ are shown in Fig. 4e. Visualizing of monolayer bands in ARPES suggests possibly a 2D electronic nature and weak van der Waals ($vdW$) interlayer coupling. Notably, surface band SS1 has the highest intensity and both SS1 / SS2 bands run across the $E_F$, which has dominant contribution into the observed SC. Figure 4e shows the Fermi contour mapping with integrated ARPES intensity. The Fermi surface (FS) along $\bar{\Gamma} - \bar{X}$ and $\bar{\Gamma} - \bar{Y}$ directions are highly identical and give a fourfold symmetry. Moreover, the FS consists of one central pocket at $\bar{\Gamma}$ point (white dashed circle, assigned to SS1), surrounded by four peripheral spots located in the middle of $\bar{\Gamma} - \bar{X}$ and $\bar{\Gamma} - \bar{Y}$ directions (assigned for the SS2). More ARPES data, e.g., multiple surface states near $E_F$ and constant energy contours (CECs) at various energies, can be found in Supplementary Figs. S8 and S9. These surface states, e.g., SS1 and SS2, observed in our ARPES measurement, implies that they could participate in unconventional SC pairing, responsible for the observed two-fold symmetric SC. A good agreement between the ARPES data and the calculated band structure for monolayer AuSn₄,

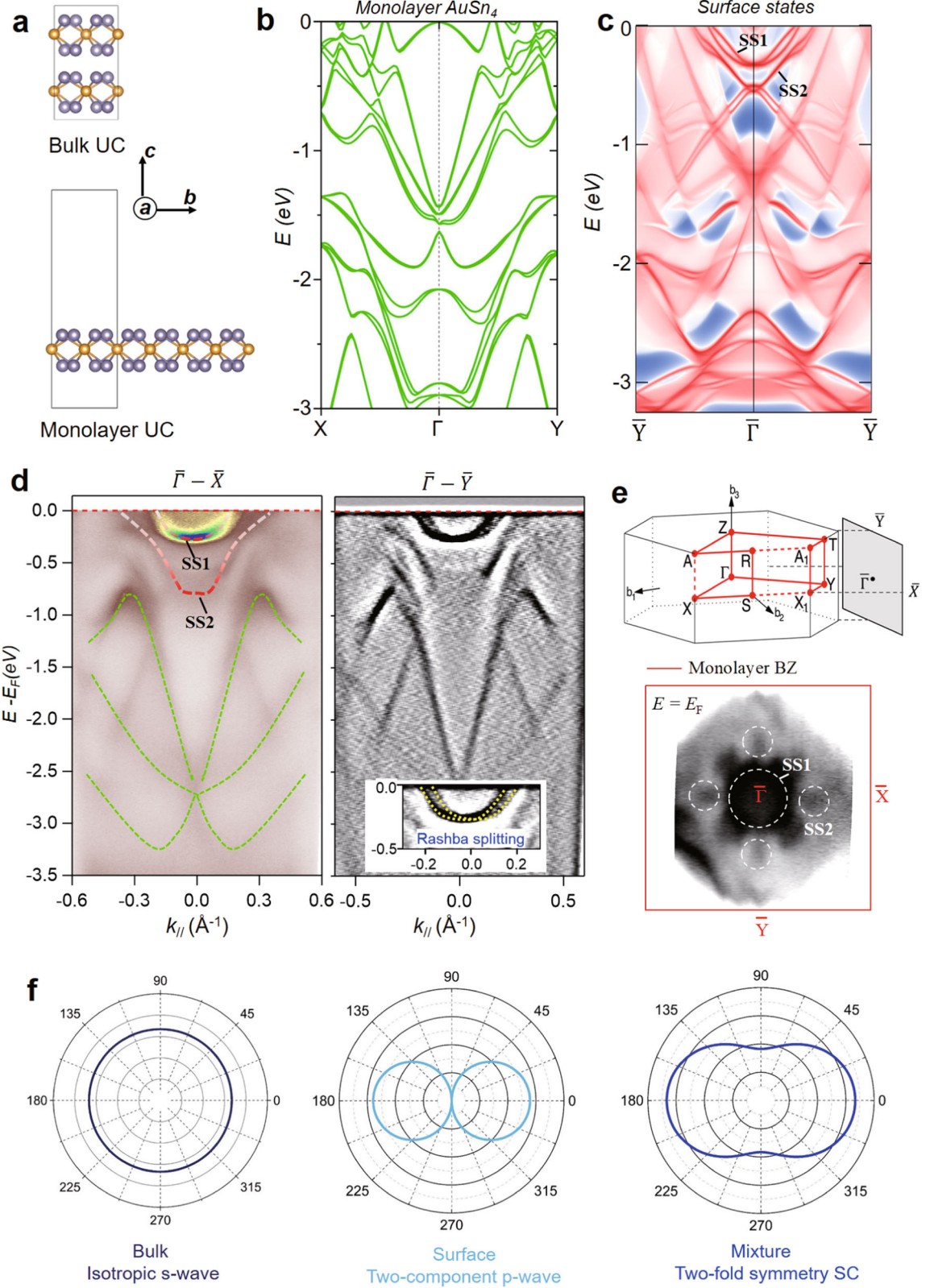

despite the difference in the binding energies, which is probably caused by the existence of long-range Coulomb in AuSn$_4$[52]. Further photon-energy-dependent ARPES (Fig. S19) suggests the 2D nature of electronic bands in AuSn$_4$; different slab calculations (Fig. S20) shows neither the slabs with various thickness nor superstructure agrees with the ARPES-measured results. All suggest that the monolayer AuSn$_4$ band structure matches the ARPES results best.

The ARPES results show that the electronic structure of AuSn$_4$ is mainly determined by the topmost AuSn$_4$ monolayer whose space symmetry can be described by point group $C_{4v}$. Due to the breaking of inversion symmetry on the surface, a typical Rashba band splitting emerges in the band structure and opposes a four-fold rotational symmetric normal state. When $T < T_c$, the superconductivity should happen in one certain irreducible representations (IR) channel, and

**Fig. 4 | Band structure, Fermi contour, and the observation of surface states.**
**a** The unit cells (UC) for the calculations of bulk and monolayer of $AuSn_4$. **b** The calculated band structures along high symmetric directions in momentum space with considering SOC effect for the monolayer of $AuSn_4$. **c** The calculated surface states from the bulk structure along $\bar{Y}$-$\bar{\Gamma}$-$\bar{Y}$ direction, where surface bands, SS1 and SS2 are marked accordingly. **d** (Left) Large energy-scaled band dispersion measured along $\bar{X}$-$\bar{\Gamma}$-$\bar{X}$ direction, where both predicted monolayer (green dashed lines) bands and surface bands SS1 and SS2 (marked as red dashed curves) are clearly

observed. (Right) the derivative plot of band dispersion measured along $\bar{Y}$-$\bar{\Gamma}$-$\bar{Y}$ direction (right) from ARPES. (Inset) Rashba splitting of surface state. **e** Fermi-surface (FS) plot of the ARPES intensity integrated within 10 meV of the chemical potential (bottom). The central circle is contributed from the SS1. Four peripheral spots (marked as white dashed circles) located along $\bar{\Gamma}$-$\bar{X}$ and $\bar{\Gamma}$-$\bar{Y}$ directions can be assigned to the SS2. (Top) Brillouin zones (BZ) of the bulk and monolayer $AuSn_4$. **f** The mixture of $p$-wave surface and $s$-wave bulk superconductivity, leads to a full-gap and two-fold symmetric superconductivity.

only the multi-dimensional channel allows multi-component super-conductivity exists and consequently breaks the rotation symmetry. Due to two-dimension limitation, the stable superconductivity order parameter $\Delta_E(\mathbf{k})$ is naturally reduced to a node $p$-wave super-conducting gap, only occurred at the surface, as shown in Fig. 4f. In the bulk, the superconducting symmetry is constrained by $D_{2h}$ point group that has no multi-dimensional IR, as the result, isotropy $s$-wave superconductivity is favored in the bulk. While the proximity effect can naturally mix two different kinds of superconductivity, the total pairing potential includes the mixture of $p$-wave surface and $s$-wave bulk superconductivity (more details in Supplementary Note 12).

## Discussion

Such mixed-SC-pairing scenario has been proposed previously, for instance, the mixing between two competing pairing instabilities, namely the $s$-wave of bulk $NbSe_2$ and an unconventional channel in few-layers in $NbSe_2$[53]. Likewise, when introducing a small orthorhom-bic distortion into the $CuO_2$ plane of cuprates, the $d$-wave pairing mixed with $s$-wave ingredient[54], gives rise to twofold in-pane critical field[55,56]. Contrast to the mixing $s$- with $d$-wave pairing in these cases, the $p$-wave pairing induced by Rashba splitting of surface bands in our $AuSn_4$ is dominant. By carefully fitting the angular dependence of resistivity at various magnetic fields (Supplementary Fig. S10), we find neither anisotropic $s$-wave nor isotropic $s$-wave and $d$-wave gap func-tions could fit the data well. These data can be fitted by the mixed $s + p$ wave gap function. All this evidence point out an unconventional $s + p$ mixed SC pairing is responsible for the observed SC in $AuSn_4$.

In summary, we fabricated high-quality $AuSn_4$ single crystals and observed a twofold symmetric superconductivity in both transport measurements and point-contact spectroscopy. The in-plane angular dependence of the resistivity at different magnetic fields below $T_c$ and the SC gap observed in STS, indicate its nature of unconventional SC. Both the $T_c$ and critical field $H_{c2}$ of $AuSn_4$ agree well in tunneling spectroscopic and transport measurements. The results of DFT cal-culation and ARPES confirm the existence of multiple surface states across $E_F$. The twofold symmetric superconductivity in $AuSn_4$ can be interpreted with the frame of the mixed $s + p$ pairing induced by Rashba splitting of surface bands.

## Methods
### Crystal growth
High-quality single crystals of $AuSn_4$ were grown by the flux method. The mixture of Au wire (99.999%) and Sn ingots (99.999%) was placed into an alumina crucible with a molar ratio of 1:8 and sealed in an evacuated quartz tube. The quartz tube was heated to 600 °C, and held there for 20 h. The temperature was then cooled to 300 °C in 1 h, and further cooled to 240 °C at a rate of 1 °C/h, and kept at 240 °C for 80 h. Finally, the excess Sn flux was removed with a centrifuge. The obtained single crystals are of thin-plate shape with typical dimen-sions of about $1 \times 1 \times 0.1$ mm$^3$. The crystal structure of as-prepared $AuSn_4$ was characterized using a Bruker diffractometer with Cu $K_\alpha$ radiation at room temperature. The energy dispersive X-ray spec-troscopy analysis of $AuSn_4$ was performed in a field emission electron microscope (Nova NanoSEM450, Czech). Spherical aberration-corrected (Cs-corrected) high angle annular dark field scanning transmission electron microscopy was performed using a FEI Titan

Themis 60–300 kV microscope equipped with a Super-X detector and operating at 200 kV.

### STM characterization
Our STM experiments were carried out on an ultrahigh vacuum (UHV) commercial STM system (Unisoku) which can reach a temperature of 400 mK by using a single-shot $^3$He cryostat. The base pressure for the experiment was $3.0 \times 10^{-10}$ Torr. $AuSn_4$ samples were cleaved in situ at 78 K and then transferred into STM. The bias voltage was applied to the samples. The STS data were obtained by a standard lock-in method that applied an additional small AC voltage with a frequency of 973.0 Hz. The $dI/dV$ spectra were collected by disrupting the feedback loop and sweeping the DC bias voltage. WSxM software was used for the post-processing of all STM data.

### Angle-resolved photoemission spectroscopy
ARPES measurements were performed with a Scienta Omicron DA30-L analyzer and monochromatized He Iα ($h\nu = 21.218$ eV) light source at School of Physics and Information Technology, Shaanxi Normal Uni-versity and Institute of Physics, Chinese Academy of Sciences. The samples were cleaved in situ and measured at 7.8 K under an ultrahigh vacuum below $6 \times 10^{-11}$Torr. The total convolved energy and angle resolutions were better than 2.5 meV and 0.2°, respectively.

### First-principles calculations
First-principles calculations were performed by DFT using the Vienna ab initio simulation package[57,58]. The plane-wave basis with an energy cutoff of 350 eV was adopted. The electron-ion interactions were modeled by the projector augmented wave potential[59] and the exchange-correlation functional was approximated by the Perdew-Burke-Ernzerhof-type generalized gradient approximation[60]. The structural relaxation for optimized lattice constants and atomic positions was performed with an energy (force) criterion of $10^{-8}$ eV (0.01 eV/Å) and by using the DFT-D3[61] method to include van der Waals corrections. Surface state calculations were performed with the WannierTools package[62], based on the tight-binding Hamilto-nians constructed from maximally localized Wannier functions (MLWF)[63].

## Data availability
All data generated or analyzed during this study are included in the published article and Supplementary Information. The data that sup-port the findings of this study are available from the corresponding authors upon request.

## Code availability
The computer code used for numerical simulations and theoretical understanding are available upon request to the corresponding author.

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

## Acknowledgements

Sample growth and TEM work was done at the School of Physics and Information Technology, Shaanxi Normal University. STM work was conducted at the School of Physics, Huazhong University of Science and Technology in Wuhan. Transport measurements were carried out at the Institute of Physics, Chinese Academy of Sciences. ARPES measurements were carried out at the Institute of Physics and Shaanxi Normal University. This work is financially supported by the National Key R&D Program of China (No. 2022YFA1403100, 2022YFA1403101, 2022YFA1403903 and 2021YFA1400403), the National Natural Science Foundation of China (No. 12204297, 12274440, 11574095, 11404175, 91745115, 21872099, 12374183 and 12004234), Innovation Program for Quantum Science and Technology (No. 2021ZD0302800), the Natural Science Foundation of Shaanxi Province (No. 2022JQ-001), the Natural Science Foundation of Henan Province (No. 232300421220), the Strategic Priority Research Program (B) of Chinese Academy of Sciences (No. XDB33010100), the Postdoctoral Innovative Talent Support Program of China (No. BX20200202), the Fundamental Research Funds for the Central Universities (No. 1301032181, GK202103023 and GK202201001), and National Major State Basic Research Development Program of China (No. 2017YFA0205000). The authors also thank Prof. Ning Hao for the helpful discussion.

## Author contributions

W.Z., R.S., J.H., Q.W., and Y.C. contributed equally to this work. W.Z. and M.P. conceived and designed the experiment. W.Z., G.L., Q.D., Y.H., J.H., Y.G., and G.C. performed transport measurements. Z.S., SL., and B.Q. performed the STM experiments at 400 mK. Q.W. and S.L. performed STM/STS measurements at 160 mK. H.J. and L.Z. performed the TEM experiments. Y.C., H.Z., X.Y., and W.Z. grew the samples. R.S., F.Z., and P.Z. carried out the DFT calculations. W.Z. and M.P. analyzed the data. J.H., W.Z., R. Zhai, J. Liu, G.Q., and T.Q. performed ARPES measurements. M.P. and W.Z. wrote the manuscript with inputs from all other authors.

## Competing interests

The authors declare no competing interests.

## Additional information

¹School of Physics and Information Technology, Shaanxi Normal University, Xi'an 710119, China. ²Science and Technology on Surface Physics and Chemistry Laboratory, Mianyang 621908, China. ³Institute of Physics and Beijing National Laboratory for Condensed Matter Physics, Chinese Academy of Sciences, Beijing 100190, China. ⁴National Laboratory of Solid State Microstructures, School of Physics, Collaborative Innovation Center of Advanced Microstructures, Nanjing University, Nanjing 210093, China. ⁵School of Physics, Huazhong University of Science and Technology, Wuhan 430074, China. ⁶Institute of Applied Physics and Computational Mathematics, Beijing 100088, China. ⁷School of Physics and Physical Engineering, Qufu Normal University, Qufu 273165, China. ⁸Shaanxi Applied Physics and Chemistry Research Institute, Xi'an 710061, China. ⁹College of Physics and Electronic Engineering, Nanyang Normal University, Nanyang 473061, China. ¹⁰School of Physical Sciences, University of Chinese Academy of Sciences, Beijing 100190, China. ¹¹Songshan Lake Materials Laboratory, Dongguan, Guangdong 523808, China. ¹²State Key Laboratory of Low-Dimensional Quantum Physics, Department of Physics, Tsinghua University, Beijing 100084, China. ¹³Beijing Academy of Quantum Information Sciences, Beijing 100193, China. ¹⁴Department of Physics, Southern University of Science and Technology, Shenzhen 518055, China. ¹⁵These authors contributed equally: Wenliang Zhu, Rui Song, Jierui Huang, Qi-Wei Wang, Yuan Cao. ✉e-mail: zhang_ping@iapcm.ac.cn; tqian@iphy.ac.cn; gfchen@iphy.ac.cn; scli@nju.edu.cn; minghupan@snnu.edu.cn; qkxue@mail.tsinghua.edu.cn

