## [Peer Review File · Nature Communications]

An Intrinsic Surface p-wave Superconductivity in Layered AuSn₄Reviewers' Comments:

Reviewer #1:

Remarks to the Author:

The searching of topological superconductors which may host Majorana quasiparticles is still a hot topic in condensed matter physics. Here the manuscript by W. Zhu et al. reported a systematical study on a possible topological superconductor of AuSn₄. The authors fabricated high-quality single crystal sample, and performed transport, point contact spectroscopy, STM/STS, ARPES measurements and DFT calculations. They observed an in-plane two-fold symmetric superconducting behavior, and a ZBCP in the point contact spectra. ARPES and DFT calculations revealed multiple surface bands crossing the E_f which has Rashba splitting. These results together suggest unconventional superconductivity which could be induced by a mixing of s + p wave pairing. I found this work interesting. Although the performed measurements are still not decisive for claiming topological superconductivity, they point out a great potential for realizing it in AuSn₄ and thus may be worthy of publication. I have some comments that need to be addressed by the authors:

1) The gap size (Δ) measured by STS in Fig. 3c is 1.5 meV, which gives a ratio of $2\Delta/kT_c \sim 14$. It is 4 times larger than the BCS ratio, but such extreme strong coupling seems to unlikely exist in such conventional metal alloy of AuSn₄. Is that any possible explanation on it? Or the gap size is over estimated due to thermo/noise broadening?

2) The SC gap in Fig. 3 only deplete 10% DOS at Fermi energy. I don't understand why the authors think this is an indication of surface state SC (page 8, line 7)? A large residue DOS at E_f usually means there exist gap nodes/nodal lines or un-gapped bands. For Fe(Te,Se), the STS measures both bulk and surface band, which give multi coherence peaks but still full gap with zero residue DOS. I think the author need to analyze the gap spectra in more detail. Finite energy resolution of the system can also result residue DOS and need to be clarified.

3) Stripe like patten is observed on the sample surface (Fig. 3a). The authors tend to ascribe them to CDW state, and Fig. S5 show some spectra indicating possible edge state. I consider these data potentially interesting. If the authors think they are really relevant to the superconductivity, they should be discussed in more detail in the main text.

4) I found no data of vortex state is presented here. Did the authors perform vortex measurement under magnetic field? Such measurement will help to clarify the origin of multi gap feature in STS, and also detects possible Majorana state in vortex core. It will significantly strengthen the conclusion of this paper.

Reviewer #2:

Remarks to the Author:

In this manuscript, the authors report magneto-transport and spectroscopy studies of a bulk superconductor AuSn₄ in combination with the DFT calculations and theoretical modellings, by focusing on the pairing symmetry in the superconducting state. Key findings of this study are (i) anisotropic superconductivity in support of its 2D or quasi-2D nature, (ii) two-fold symmetric in-plane angular dependence of magnetoresistance below the upper critical field which does not match the symmetry of bulk crystal (four-fold), (iii) emergence of zero-bias peak in the point-contact spectrum in support of the unconventional (topological) nature of superconductivity, (iv) V-shaped superconducting gap which is not compatible with the isotropic s-wave symmetry, and (v) observation of metallic Rashba-spin-split surface state which would play a key role to the superconductivity at the surface. By comprehensively taking into account these observations, they have concluded surface p-wave and bulk s-wave nature of superconductivity in AuSn₄. If the p-wave superconductivity is indeed realized, AuSn₄ would be a useful platform to search for Majorana modes, as the authors have suggested. Overall, experiments are carefully carried out, the manuscript is well written, introduction is accessible for general readers, and their interpretation is mostly straightforward. However, I find a few issues which need to be clarified to support intrinsic topological p-wave superconductivity in this compound, as listed below, before I

recommend the publication of this manuscript in Nature Communications.

1) Although the authors have clearly shown superstructure with 1D-type CDW-like modulation from their STM measurements, influence of this superstructure to the band structure and pairing symmetry need to be further elaborated. For example, why the ARPES data show four-fold-symmetric Fermi-surface pattern despite the presence of anisotropic (1D-like) superstructure? Could it be due to the presence of two surface domains? Or, Was the ARPES measurement carried out above the CDW transition temperature? (in this case, please specify the critical temperature). What is the relationship between the two-fold symmetric superlattice and pairing symmetry? Can the superconducting gap be affected by this superstructure?

2) While the authors have suggested a good agreement between the ARPES data and the calculated band structure for monolayer AuSn₄, there exist several quantitative differences. For example, the V-shaped band seen at the binding energy of 0.7-2.5 eV in the experiment (Fig. 4d) moves significantly upward in the calculation (0.4-1.5 eV). Complicated band dispersion around the X and Y points near EF in the calculation is missing in the experiment. It is unclear to me to what extent monolayer AuSn₄ is a good starting point to understand the experimental band dispersion. It is also unclear whether the monolayer-derived bands insisted by the authors show 2D dispersion. In this regard, it would be useful to carry out photon-energy-dependent ARPES measurements on AuSn₄ and compare the result with the calculated bulk band structure, before comparing with the calculation for monolayer AuSn₄. In relation to my comment 1), perhaps slab calculations that incorporate the superstructure can be compared with the ARPES data.

3) If the p-wave topological superconductivity indeed occurs at the surface, dispersive Majorana edge mode is expected to appear at the edge (in the AuSn₄ case, it would correspond to the atomic steps of the cleaved surface). Can the authors access such an edge state?

4) Most of the recent STM studies claimed topological superconductivity based on the observation of zero-energy mode at the vortex cores or at the corner of 1D wires. On the other hand, this evidence is missing in this study. Although the authors have pointed out the observation of Majorana zero mode as a future challenge, I think that the authors can carry out this experiment because they can cool down to 400 mK which is sufficiently lower than T_c. If there are some other obstacles to observe the Majorana modes, please state explicitly.

Reviewer #3:

Remarks to the Author:

The manuscript submitted by Wenliang Zhu et al. reports a set of studies on the layered superconductor AuSn₄, including angular dependent transport, point-contact Andreev reflection spectroscopy, scanning tunneling microscopy/spectroscopy, angle-resolved photoemission spectroscopy, and first-principle calculation. The authors claim that their results support the presence of intrinsic surface p-wave superconductivity in the material.

Although the authors have carried out extensive studies on the two-dimensional superconductor using various techniques and the results are certainly helpful in understanding of the superconductivity of the material, the manuscript contains fundamental flaws and the conclusions are highly debatable. Therefore, in my opinion, the present manuscript does not reach the standard of Nature Communications. My detailed comments are attached below.

1. The authors presented an angular-dependent magnetoresistance study of this type-II superconductor in various in-plane magnetic fields below the H_{c2}// - this is very confusing because it is against the knowledge of type-II superconductivity. A type-II superconductor remains superconducting in a magnetic field below H_{c2}. So, this raises a question regarding how the authors define the H_{c2} at in this study a given temperature, but the definition is not explicitly explained in the manuscript. I assume that the authors take the onset of the resistivity drop as the field-temperature relation for the H_{c2}(T), but this is not the conventional definition for a type-II superconductor. I believe, with a lot of respect, that the senior co-authors of the manuscript should have recognized this fundamental issue earlier.

2. Continuing the comment above, the angular-dependent magnetoresistance measurements were actually performed in a field range where pairing fluctuation reduces the resistance (but not forming any superconducting path all the way through the material). The authors define the angle between the field direction and the crystal axis a as θ and show in Fig. 2e that the angular dependence of the magnetoresistance when $I//b$ is 90 degrees rotated from that when $I//a$. But I don't understand why the case was made so complicated because if θ is simply defined as the angle between current and field (that is, regardless of the crystal lattice), the two angular dependences shown in Fig. 2e will be identical. Therefore, the "two-fold" angular dependence entirely depends on the relative angle between current and field. In my opinion, given the field range where the two-fold angle-dependent MR is observed, the in-plane anisotropy can be explained by the interplay of magnetic flux and pairing fluctuation. For instance, magnetic flux suppresses pairing fluctuation, but the strength/level of the suppression changes with the angle between the field and the current. In fact, it would be interesting to see that at a given in-plane field, whether the zero-resistance temperature is related to the angle between current and field.

3. There is a long history of debate on whether a zero-bias conductance peak (ZBCP) can be properly related to unconventional superconductivity in point-contact Andreev reflection measurements. For example, Gifford et al. [J. Appl. Phys. 120, 163901 (2016)] has a detailed study of ZBCP observed in point-contact spectroscopy measurements. The present manuscript suggests that a two-fold angular dependence of the ZBCP at a field of 120 Oe where AuSn₄ remains superconducting is a signature of unconventional Andreev bound state. However, similar to Comment 2 described above, the in-gap features may change due to the variation of the interaction between the penetrated magnetic flux (assuming the external magnetic field is above H_{c1}) and the paired electrons. Given the device design shown in the inset of Fig. S4, this possibility cannot be ruled out in the present study. Also, it is very unusual to relate a pair of dips (± 3 mV), instead of peaks, to superconductivity or Andreev bound states in point-contact measurements. In addition, I also notice that the contact conductance increased by only $\sim 4\%$ at zero bias. Although one may find some reports having similar "enhancement" in literature, this is not an ideal Andreev-reflection junction. In such a case, many artificial/accidental features may be exaggerated, and analysis should not rely on such small features.

4. The STS data shown in Fig. 3 indicate a gap-like feature (called "outer gap" in the text) at ~ 3.2 mV in addition to the smaller gap feature at ~ 1.5 mV, and the authors claim it as evidence of multi bands, presumably referring to multi-band superconductivity. However, the "outer gap" is very weak and cannot be distinguished from the "pseudogap-like" feature that survives far above the T_c or H_c of the material. Therefore, I feel that the conclusion of multiband lacks sufficient support in the STS measurements. In addition, the zero-bias conductance of the tunneling spectrum is only about 8% smaller than the normal conductance, and in such a case, scattering-induced spectrum broadening can introduce a significantly large error for the fitting parameters. I notice that although the fitting curves in Fig. 3c and Fig. S11 d have similar "shape", the magnitudes of the dips are completely different. The authors should list all the fitting parameters in the manuscript.

Manuscript NCOMMS-23-24487-T

REVIEWER COMMENTS

Reviewer #1 (Remarks to the Author):

The searching of topological superconductors which may host Majorana quasiparticles is still a hot topic in condensed matter physics. Here the manuscript by W. Zhu et al. reported a systematical study on a possible topological superconductor of AuSn₄. The authors fabricated high-quality single crystal sample, and performed transport, point contact spectroscopy, STM/STS, ARPES measurements and DFT calculations. They observed an in-plane two-fold symmetric superconducting behavior, and a ZBCP in the point contact spectra. ARPES and DFT calculations revealed multiple surface bands crossing the E_F which has Rashba splitting. These results together suggest unconventional superconductivity which could be induced by a mixing of $s + p$ wave pairing. I found this work interesting. Although the performed measurements are still not decisive for claiming topological superconductivity, they point out a great potential for realizing it in AuSn₄ and thus may be worthy of publication. I have some comments that need to be addressed by the authors:

Authors' reply: We thank the referee for the positive evaluation and recommendation of our work. Below, we address the comments raised by this referee point-to-point and revise the manuscript accordingly.

Com. 1. The gap size (Δ) measured by STS in Fig. 3c is 1.5 meV, which gives a ratio of $2\Delta/k_B T_c \sim 14$. It is 4 times larger than the BCS ratio, but such extreme strong coupling seems to unlikely exist in such conventional metal alloy of AuSn₄. Is that any possible explanation on it? Or the gap size is over estimated due to thermo/noise broadening?

Authors' reply: We thank the reviewer for his/her constructive comment. Actually, thermal broadening effect could induce about $4k_B T$ (k_B is Boltzmann constant) broadening on STS measurement (composed of $\sim 2k_B T$ for the sample and $\sim 2k_B T$ for the tip) [Jennifer E Hoffman, *Rep. Prog. Phys.* **74**, 124513 (2011)], affects both the size

and the depth of SC gap. At the temperature of 400 mK, $4k_B T$ is about 0.138 meV. Another effect to cause a large gap size, could be from the large lock-in modulation bias V_{mod} , which we used in previous experiment is about 300 μ V.

In order to eliminate the thermal broadening effect, we carried out the dI/dV measurements at a lower temperature \sim 160 mK, by collaborated with Prof. Shaochun Li in Nanjing University, China. In **Fig. R1**, a SC gap is clearly observed on the flat terrace of AuSn₄ surface at 160 mK. The gap size is about 0.487 meV, with the depth of gap about 87%. Note, we adopt much smaller modulation bias as $V_{mod} = 40 \mu$ V. Now the value of $2\Delta/k_B T_c$ is about \sim 4.7, which is in the range of the medium coupling for BCS ratio.

Fig. R1. A representative dI/dV spectra taken at 160 mK and 0 Oe. The dI/dV curve measured on most of terraces. ($V_b = 1.35$ mV, $I_t = 200$ pA, $V_{mod} = 40 \mu$ V).

In the revised manuscript, we update the Fig. 3c with new STS and revise the related content.

Fig. R2. The updated Fig.3

In page 8 line 11, we revise to “A typical normalized dI/dV spectrum measured at 0.16 K (**Fig. 3c**) shows a superconductive gap feature, a deep V-shaped gap about $\Delta \sim 0.487$ meV (defined as half of the gap width) with the depth of gap about 87%.”

In page 8 line 20, we revise to “The ratio $2\Delta(0)/k_B T_c$ (k_B is the Boltzmann constant) is estimated to be ~ 4.7 , which is in the range of the medium coupling Bardeen–Cooper–Schrieffer (BCS) superconductors.”

In the caption of Fig. 3, we rewrite “A representative dI/dV spectrum taken at 0.16 K, showing a superconducting gap with the size and depth of gap about 0.487 meV and 87%, respectively. Here, we adopt small lock-in modulation bias (V_{mod}) ~ 40 μ V in order to eliminate the broadening effect.” And add “Note, the STS taken at 0.38 K with 0.3 mV V_{mod} , give a dual-gap feature with enlarged gap size and shallower depth, probably due to the thermal broadening effect and large lock-in modulation bias.”

Com. 2. The SC gap in Fig. 3 only deplete 10% DOS at Fermi energy. I don't understand why the authors think this is an indication of surface state SC (page 8, line 7)? A large residue DOS at E_F usually means there exist gap nodes/nodal lines or un-gapped bands. For Fe (Te, Se), the STS measures both bulk and surface band, which give multi coherence peaks but still full gap with zero residue DOS. I think the author need to analyze the gap spectra in more detail. Finite energy resolution of the system can also result residue DOS and need to be clarified.

Authors' reply: We thank the reviewer for this constructive suggestion. With newly-measured STS data presented in **Fig. R1**, now the SC gap deplete 87% DOS at Fermi energy. Previous large residue DOS at E_F seems to be caused by thermal broadening effect and a large lock-in modulation bias V_{mod} we used. Now there is still a 13% residue DOS at E_F , which may mean there exists gap nodes/nodal lines or un-gapped bands, as the reviewer suggested.

In page 8 line 25, we revise to “It is worth to note that there is still a 13% residue DOS at E_F , which may indicate there exists gap nodes/nodal lines or un-gapped bands.” and remove ~~“the depth of superconducting gap is only 87% of normalized density of states, while the transport results show a 100% superconducting volume ratio. Such discrepancy may indicate that the observed gap is originated from the surface states SC, similar to the case of $\text{FeTe}_{0.55}\text{Se}_{0.45}$, in which the SC is induced in the surface bands through interband scattering^[47]. Possibly, the bulk bands SC is pushed underneath the surface layer and hardly detected by STS.”~~.

Com. 3. Stripe like pattern is observed on the sample surface (Fig. 3a). The authors tend to ascribe them to CDW state, and Fig. S5 show some spectra indicating possible edge state. I consider these data potentially interesting. If the authors think they are really relevant to the superconductivity, they should be discussed in more detail in the main text.

Authors' reply: We agree with the reviewer and carry out more investigation on the stripe-like pattern on the surface. In Fig. R2, we carried out STS measurements at 160 mK. As we mentioned, there are two different modulations on this surface, $\sqrt{2} \times 2\sqrt{2}$ superstructure (CDW) and stripe-like pattern. As shown in Fig. R2, both line STS surveys, show clearly the SC gap modulated by both $\sqrt{2} \times 2\sqrt{2}$ superstructure (CDW) and stripe pattern.

Recently, pair density wave (PDW) has been found in a number of superconductors, such as $\text{EuRbFe}_4\text{As}_4$ [*Nature* **618**, 940 (2023)], $\text{Fe}(\text{Te},\text{Se})$ films [*Nature* **618**, 934 (2023)], UTe_2 [*Nature* **618**, 928 (2023), *Nature* **618**, 921 (2023)]. A PDW state is an exotic phase of matter, a Larkin-Ovchinnikov-like state [Larkin, A. I. and Ovchinnikov, Y. I. Inhomogeneous state of superconductors. *Sov. Phys. JETP* **20**, (1965); Agterberg, D.F. *et al*, The physics of pair-density waves: Cuprate superconductors and beyond. *Annual Review of Condensed Matter Physics* **11**, (2020)] with finite momentum Cooper pairs. Actually, the Ginzburg-Landau theory predicted a system with coexisting uniform triplet superconductivity and triplet PDWs, leads to a daughter CDW order, with the same wavevector as the PDW, as previously established. Our new results suggest that the observed CDW state is relevant to the superconductivity, which may point out the nature of the coexisting triplet superconductivity and PDWs.

Fig. R2. The correlations between SC gap and CDW modulations. **a**, Topographic image (I_t : 100 pA, V_b : 1.0 V, and image size: $20 \times 20 \text{ nm}^2$) taken at temperature of 4.2K. **b-c**, The line dI/dV spectroscopic survey taken along the blue (black) arrows in panel **a**. Inset: intensity map around the coherence peak energy, clearly showing the

modulation of the coherence peaks in energy. The STS set-up conditions: **b**: $V_b = 2.7$ mV, $I_t = 200$ pA, $V_{mod} = 40$ μ V, **c**: $V_b = 1.35$ mV, $I_t = 200$ pA, $V_{mod} = 27$ μ V taken at the temperature of 160 mK.

In page 9 line 7, we add “Our STS measurements taken at 160 mK, show clearly the SC gaps are modulated with both $\sqrt{2} \times 2\sqrt{2}$ superstructure (CDW) and stripe pattern (Supplementary Fig. S16), suggesting that the observed CDW state is relevant to the superconductivity.”.

We add the discussion and Fig. R2 into SI as Note 14 and Fig. S16, respectively.

Com. 4. I found no data of vortex state is presented here. Did the authors perform vortex measurement under magnetic field? Such measurement will help to clarify the origin of multi gap feature in STS, and also detects possible Majorana state in vortex core. It will significantly strengthen the conclusion of this paper.

Authors' reply: We thank the reviewer for this constructive suggestion. By collaborated with Prof. Shaochun Li in Nanjing University, China, we perform the vortex measurements. As shown in Fig. R3, the field dependence of SC gap (Fig. R3a) are measured at 160 mK and various fields, e.g. 20, 96, 134 and 173 Oe. The SC gaps are gradually filled with increasing the fields and become relatively-shallow at 173 Oe. However, the vortex measurement (zero-bias-conductance (ZBC) map) doesn't show the existence of vortex at both 20 Oe (Fig. R3b) and 173 Oe (Fig. R3c). Using the formula of $\xi_{ab} = \xi_c H_{c2}^{ab} / H_{c2}^c$, we calculated the value of the vortex core radius $r_c \approx \xi_{ab}$ to be ~ 190.8 nm [*Phys. Rev. B* **100**, 064516 (2019), *Phys. Rev. B* **79**, 134526 (2009)]. Here, H_{c2}^{ab} and H_{c2}^c are the upper critical field for $H//ab$ and $H//c$, respectively, and ξ_{ab} and ξ_c are the coherence length for $H//ab$ and $H//c$, respectively. For AuSn₄, $H_{c2}^{ab} = 1621$ Oe, $H_{c2}^c = 643$ Oe, and $\xi_c = 71.69$ nm. The distance between the center of the vortex lattice should be less than 410.73 nm estimated with the formula of $a = \sqrt{\frac{\phi_0}{B} d}$ [*Dokl. Acad. Nauk.* **86**, 489 (1952), *Dokl. Acad. Nauk.* **86**, 501 (1952), *J. Phys. Chem. Solids* **2**, 199 (1957)]. Here, d is a constant coefficient and B is the lower critical field. For AuSn₄, $d = 1$ and $B = 118$ Oe. We find that r_c is roughly comparable with

$a/2$. Therefore, the reason for the absence of vortices in a superconductor, which may be due to the overlap of vortex [Tinkham, M. Introduction to superconductivity. Courier Corporation, (2004)].

Although we didn't observe the vortices under the fields, however, the point dI/dV spectrum under the fields indeed show some in-gap DOS peaks (the curves of 96 Oe and 134 Oe in **Fig. R3a**), which may be related to possible Majorana state near the vortex core.

Fig. R3. Gap evolution and zero-bias maps under various fields. **a**, Point spectra under various fields, e.g. 20, 96, 134 and 173 Oe ($V_b = 1.35$ mV, $I_t = 200$ pA, $V_{mod} = 40$ μ V). **b** Topographic image (I_t : 100 pA, V_b : 6.75 mV, and image size: 185×185 nm²) and simultaneously measured ZBC map ($V_{mod} = 675$ μ V) under 20 Oe. **c**, Topographic image (I_t : 100 pA, V_b : 1.35 mV, and image size: 55×55 nm²) and simultaneously measured ZBC map ($V_{mod} = 135$ μ V) under 173 Oe.

In page 8 line 27, we add the discussion “Furthermore, we perform vortex measurement under magnetic fields, however, zero-bias-conductance (ZBC) maps do not show the existence of vortex, which may be due to the overlaps of vortex [51] (see the details of analysis in Supplementary Note 18).”

We add the reference:

51. Tinkham, M. Introduction to superconductivity. Courier Corporation, (2004).

We add the discussion into SI as **Note 18**.

We add the references into SI as following:

8. Chen, D. Y. *et al.* Superconducting properties in a candidate topological nodal line semimetal SnTaS₂ with a centrosymmetric crystal structure. *Phys. Rev. B* **100**, 064516 (2019).
9. Brandt, E. H. Vortex-vortex interaction in thin superconducting films. *Phys. Rev. B* **79**, 134526 (2009).
10. Abrikosov, A. A. An influence of the size on the critical field for type II superconductors. *Dokl. Akad. Nauk.* **86**, 489 (1952).
11. Zavaritskii, N. Superconducting properties of thallium and tin films condensed at low temperatures. *Dokl. Acad. Nauk.* **86**, 501 (1952).
12. Abrikosov, A. A. The magnetic properties of superconducting alloys. *J. Phys. Chem. Solids* **2**, 199 (1957).
13. Tinkham, M. Introduction to superconductivity. Courier Corporation, (2004).

Reviewer #2 (Remarks to the Author):

In this manuscript, the authors report magneto-transport and spectroscopy studies of a bulk superconductor AuSn_4 in combination with the DFT calculations and theoretical modellings, by focusing on the pairing symmetry in the superconducting state. Key findings of this study are (i) anisotropic superconductivity in support of its 2D or quasi-2D nature, (ii) two-fold symmetric in-plane angular dependence of magnetoresistance below the upper critical field which does not match the symmetry of bulk crystal (four-fold), (iii) emergence of zero-bias peak in the point-contact spectrum in support of the unconventional (topological) nature of superconductivity, (iv) V-shaped superconducting gap which is not compatible with the isotropic s-wave symmetry, and (v) observation of metallic Rashba-spin-split surface state which would play a key role to the superconductivity at the surface. By comprehensively taking into account these observations, they have concluded surface p -wave and bulk s -wave nature of superconductivity in AuSn_4 . If the p -wave superconductivity is indeed realized, AuSn_4 would be a useful platform to search for Majorana modes, as the authors have suggested. Overall, experiments are carefully carried out, the manuscript is well written, introduction is accessible for general readers, and their interpretation is mostly straightforward. However, I find a few issues which need to be clarified to support intrinsic topological p -wave superconductivity in this compound, as listed below, before I recommend the publication of this manuscript in Nature Communications.

Authors' reply: We thank the referee for the positive comments and for elegantly summarizing the key results in this work. Below, we address the comments raised by this referee point-to-point and revise the manuscript accordingly.

Com. 1. Although the authors have clearly shown superstructure with 1D-type CDW-like modulation from their STM measurements, influence of this superstructure to the band structure and pairing symmetry need to be further elaborated. For example, why the ARPES data show four-fold-symmetric Fermi-surface pattern despite the presence of anisotropic (1D-like) superstructure? Could it be due to the presence of two surface

domains? Or, Was the ARPES measurement carried out above the CDW transition temperature? (in this case, please specify the critical temperature). What is the relationship between the two-fold symmetric superlattice and pairing symmetry? Can the superconducting gap be affected by this superstructure?

Authors' reply: We thank the referee for this constructive comment. Actually, STM images show the directions of CDW and Q1D modulations can be along either *a* or *b* axis (SI, Note 8), leads to four-fold-symmetric Fermi-surface pattern in ARPES, which will not introduce the two-fold anisotropic SC in macroscopic scale. As shown in **Fig. S6**, the full edge is with step height $\sim 11 \text{ \AA}$ and the direction of CDWs (defined as the direction of $\sqrt{2} \times 2\sqrt{2}$) in the upper and lower terraces are perpendicular to each other (marked as red dashed arrows). All these results suggest that CDWs can be developed along both *a* and *b* directions, which cannot provide for the underlying mechanism of two-fold symmetry of SC.

ARPES measurement was carried out at 6 K, while this CDW transition temperature is hardly determined based on our STS data. In **Fig. 3f**, we measured variable-temperature STS from 5 K to 10 K above the T_c , a pseudogap-like feature develops above T_c and persists up to 10 K, which may be corresponding to the existence of observed CDW gap above 10 K.

For the relationship between the superlattice and pairing symmetry, In **Fig. R2**, we carried out STS measurements at 160 mK. As we mentioned, there are two different modulations on this surface, $\sqrt{2} \times 2\sqrt{2}$ superstructure (CDW) and stripe-like pattern. As shown in **Fig. R2**, both line STS surveys, show clearly the SC gap modulations with both $\sqrt{2} \times 2\sqrt{2}$ superstructure (CDW) and stripe pattern. Our new results suggest that the observed CDW state is relevant to the superconductivity, which may point out the nature of the coexisting triplet superconductivity and PDWs.

Fig. R2. The correlations between SC gap and CDW modulations. **a**, Topographic image (I_t : 100 pA, V_b : 1.0 V, and image size: $20 \times 20 \text{ nm}^2$) taken at temperature of 4.2 K. **b-c**, the line dI/dV spectroscopic survey taken along the blue (black) arrows in panel **a**. Inset: intensity map around the coherence peak energy, clearly showing the modulation of the coherence peaks in energy. The STS set-up conditions: **b**: $V_b = 2.7 \text{ mV}$, $I_t = 200 \text{ pA}$, $V_{mod} = 40 \text{ } \mu\text{V}$, **c**: $V_b = 1.35 \text{ mV}$, $I_t = 200 \text{ pA}$, $V_{mod} = 27 \text{ } \mu\text{V}$ taken at the temperature of 160 mK.

In page 9 line 7, we add “Our STS measurements taken at 160 mK, show clearly the SC gaps are modulated with both $\sqrt{2} \times 2\sqrt{2}$ superstructure (CDW) and stripe pattern (Supplementary Fig. S16), suggesting that the observed CDW state is relevant to the superconductivity.”.

We add the discussion and Fig. R2 into SI as Note 14 and Fig. S16, respectively.

Com. 2. While the authors have suggested a good agreement between the ARPES data and the calculated band structure for monolayer AuSn_4 , there exist several quantitative differences. For example, the V-shaped band seen at the binding energy of 0.7-2.5 eV in the experiment (Fig. 4d) moves significantly upward in the calculation (0.4-1.5 eV). Complicated band dispersion around the X and Y points near E_F in the calculation is missing in the experiment. It is unclear to me to what extent monolayer AuSn_4 is a good starting point to understand the experimental band dispersion. It is also unclear whether the monolayer-derived bands insisted by the authors show 2D dispersion. In this regard, it would be useful to carry out photon-energy-dependent

ARPES measurements on AuSn₄ and compare the result with the calculated bulk band structure, before comparing with the calculation for monolayer AuSn₄. In relation to my comment 1), perhaps slab calculations that incorporate the superstructure can be compared with the ARPES data.

Authors' reply: Thanks for the insightful suggestion and we agree with the referee that the k_z dispersion and slab calculations are significant to understand the ARPES data. According to this comment, we have given the detailed replies as the following three points.

1) First, as shown in **Fig. R4**, different slab models were adopted to simulate the real condition. The corresponding band structure shows a good metallicity, however, none of them agrees with the ARPES-measured results. As suggested by the referee, to simulate the surface superstructure, we also construct a $2\sqrt{2} \times \sqrt{2}$ supercell slab which is in consistent with the STM observation and the corresponding band structure is shown in **Fig. R4d**. Obviously, it deviates the ARPES measurements a lot, which indicates that the ARPES results cannot be interpreted by surface superstructure. The phonon band of bulk AuSn₄ shown in **Fig. R4f** and **R4g** indicates an evident dynamic stability of this system, which demonstrate that the superstructure on the surface (CDW-like state) is not originated from crystal lattice instability. As mentioned above, STM images show the directions of CDW and Q1D modulations can be along either a or b axis, leads to four-fold-symmetric Fermi-surface pattern in ARPES, which cannot provide the underlying mechanism for two-fold symmetry of SC. In conclusion, the slab model cannot capture the dispersion relationship nor the Fermi surface's symmetry detected by ARPES, on the contrary, monolayer model fits it much better.

Fig. R4. Band structure for slabs of different thickness. **a**, $n=2$; **b**, $n=4$; **c**, $n=6$; **e**, $n=12$, where n is the number of layers. **d**, slab calculations that incorporate the $2\sqrt{2} \times \sqrt{2}$ superstructure. **f** and **g** are the phonon spectrum of bulk and monolayer AuSn₄, respectively.

2) According to the suggestion, photon-energy-dependent ARPES measurements had been performed at the Dreamline beamline of the Shanghai Synchrotron Radiation Facility with a Scienta Omicron DA30L analyzer. We show the data of k_z dispersion in **Fig. R5**, in which photon-energy-dependent band dispersion is almost a series of vertical lines, suggesting that there is no obvious k_z dispersion and the coupling between two single layers is very weak. So the motion of electrons in AuSn₄ is restricted in each 2D single layer, especially for the surface electrons. Thus, we think that the monolayer AuSn₄ is a good starting point to understand the experimental band

dispersion, and judging from the calculations, it is the monolayer band structure matches experiment best. Besides, we find that the photon energy of about 22 eV is a good choice for ARPES studies of AuSn₄.

Fig. R5. Photon-energy-dependent ARPES measurements show that there is no obvious k_z dispersion.

3) In order to clarify that complicated band dispersion around the X and Y points near E_F in the calculation is missing in the experiment, we have re-adjusted the color scale of the marked areas of **Fig. R6a** in attempt to better distinguish band dispersion. As shown in **Fig. R6b**, the observed bands (marked by red/green dashed lines in **a** and **b**) can be attributed to the calculations of the monolayer (**c**) and surface states (**d**), despite the difference in the binding energies.

4) Such difference that the ARPES data displays a broadening of the bandwidth with respect to the predictions of DFT calculations, is different from the renormalization induced by underestimating quasiparticle masses and overestimating bandwidths. The so-called “inverse band renormalization” could be due to the long-range Coulomb interaction. As mentioned by the referee, the energy width of V-shaped band is about

1.1 eV in DFT results, however it is about 1.8 eV in experiment, which is broadened near 60% than DFT. The similar phenomenon has been observed in the references [*Phys. Rev. Lett.* **121**, 117002 (2018)], where ARPES-measured bandwidth of $\text{Ba}_{0.51}\text{K}_{0.49}\text{BiO}_3$ is about 50% larger than DFT calculations involved with GGA exchange-correlation functional. Back to our system, the electrons of AuSn_4 have a very strong two-dimensional characteristic. It is known that dielectric screening in 2D is much smaller than it in 3D so that the long-range Coulomb interaction is much easier to survive in (quasi-) 2D electron system. Based on the above analysis, we think the difference of bandwidth between DFT and ARPES is probably caused by the existence of long-range Coulomb in AuSn_4 .

To make this point more clearly for the readers, we have added the data of k_z dispersion, slab calculations and the corresponding discussions in SI (marked in red).

In page 10 line 12, we add “A good agreement between the ARPES data and the calculated band structure for monolayer AuSn_4 , despite the difference of the binding energies, which is probably caused by the existence of long-range Coulomb in AuSn_4 [53]. Further photon-energy-dependent ARPES (**Fig. S19**) suggests the 2D nature of electronic bands in AuSn_4 ; different slab calculations (**Fig. S20**) show neither the slabs with various thickness nor superstructure agrees with the ARPES-measured results. All suggests that the monolayer AuSn_4 band structure matches the ARPES results best.”.

We add the reference:

53. Wen, C. H. P. *et al.* Unveiling the Superconducting Mechanism of $\text{Ba}_{0.51}\text{K}_{0.49}\text{BiO}_3$, *Phys. Rev. Lett.* **121**, 117002 (2018).

Fig. R6. Experimental ARPES result (a and b) with the comparison of calculations (c and d). The red (green) dashed lines in a and b are the bands deduced from the calculated bands of the monolayer (surface states). b, band dispersion in a marked in blue dashed lines.

Com. 3. If the p -wave topological superconductivity indeed occurs at the surface,

dispersive Majorana edge mode is expected to appear at the edge (in the AuSn₄ case, it would correspond to the atomic steps of the cleaved surface). Can the authors access such an edge state?

Authors' reply: We thank the referee for this constructive comment. We performed the STS measurements on the flat terrace and at the edge at 160 mK. In **Fig. R7**, the SC gap measured at the edge (blue) shows a slightly smaller gap size and shallower gap depth, both suggesting the possible existence of dispersive Majorana edge mode at the edge.

Fig. R7. STS measured on the terrace and at the edge. **a-b**, Topographic images showing the locations of point dI/dV spectra (**a**: I_t : 100 pA, V_b : 100 mV, and image size: 130×130 nm², **b**: I_t : 100 pA, V_b : 0.675 V, and image size: 130×130 nm²). **c**, the comparison of dI/dV spectra on the terrace (red) and at the edge (blue). The STS set-up conditions: edge: $V_b = 6.75$ mV, $I_t = 100$ pA, $V_{mod} = 67.5$ μ V, terrace: $V_b = 1.35$ mV, $I_t = 300$ pA, $V_{mod} = 27$ μ V, taken at the temperature of 160 mK and magnetic field of 20 Oe.

In page 9 line 10, we add a sentence "Interestingly, STS measurements on the flat terrace and at the edge at 160 mK (**Supplementary Fig. S15**), show that a slightly smaller size and shallower depth of SC gap measured at the edge, comparing to the gap measured on the flat terrace, suggesting the possible existence of dispersive Majorana edge mode [52].".

We add the reference:

52. A. Palacio-Morales, E. Mascot, S. Cocklin, H. Kim, S. Rachel, D. K. Morr, R. Wiesendanger, Atomic-scale interface engineering of Majorana edge modes in a 2D magnet-superconductor hybrid system. *Sci. Adv.* **5**, eaav6600 (2019).

In SI, we add it as **Note 13** and Supplementary **Fig. S15**.

Com. 4. Most of the recent STM studies claimed topological superconductivity based on the observation of zero-energy mode at the vortex cores or at the corner of 1D wires. On the other hand, this evidence is missing in this study. Although the authors have pointed out the observation of Majorana zero mode as a future challenge, I think that the authors can carry out this experiment because they can cool down to 400 mK which is sufficiently lower than T_c . If there are some other obstacles to observe the Majorana modes, please state explicitly.

Authors' reply: We thank the referee for this constructive comment. By collaborating with Prof. Shaochun Li in Nanjing University, China, we perform the vortex measurements. As shown in **Fig. R3**, the field dependence of SC gap (**Fig. R3a**) are measured at 160 mK and various fields, e.g. 20, 96, 134 and 173 Oe. The SC gaps are gradually filled with increasing the fields and become relatively-shallow at 173 Oe. However, the vortex measurement (zero-bias dI/dV mapping) can't show the existence of vortex at both 20 Oe (**Fig. R3b**) and 173 Oe (**Fig. R3c**). Using the formula of $\xi_{ab} = \xi_c H_{c2}^{ab} / H_{c2}^c$, we calculated the value of the vortex core radius $r_c \approx \xi_{ab}$ to be ~ 190.8 nm [*Phys. Rev. B* **100**, 064516 (2019), *Phys. Rev. B* **79**, 134526 (2009)]. Here, H_{c2}^{ab} and H_{c2}^c are the upper critical field for $H//ab$ and $H//c$, respectively, and ξ_{ab} and ξ_c are the coherence length for $H//ab$ and $H//c$, respectively. For AuSn₄, $H_{c2}^{ab} = 1621$ Oe, $H_{c2}^c = 643$ Oe, and $\xi_c = 71.69$ nm. The distance between the center of the vortex lattice should be less than 410.73 nm estimated with the formula of $a = \sqrt{\frac{\Phi_0}{B}} d$ [*Dokl. Acad. Nauk.* **86**, 489 (1952), *Dokl. Acad. Nauk.* **86**, 501 (1952), *J. Phys. Chem. Solids* **2**, 199 (1957)]. Here, d is a constant coefficient and B is the lower critical field. For AuSn₄,

$d = 1$ and $B = 118$ Oe. We find that r_c is roughly comparable with $a/2$. Therefore, the reason for the absence of vortices in a superconductor, which may be due to the overlap of vortex [Tinkham, M. Introduction to superconductivity. Courier Corporation, (2004)].

Although we didn't observe the vortices under the fields, however, the point dI/dV spectrum under the fields indeed show some in-gap DOS peaks (the curves of 96 Oe and 134 Oe in Fig. R3a), which may be related to possible Majorana state near the vortex core.

Fig. R3. Gap evolution and zero-bias maps under various fields. **a**, Point spectra under various fields, e.g. 20, 96, 134 and 173 Oe ($V_b = 1.35$ mV, $I_t = 200$ pA, $V_{mod} = 40$ μ V). **b** Topographic image (I_t : 100 pA, V_b : 6.75 mV, and image size: 185×185 nm²) and simultaneously measured zero-bias dI/dV map ($V_{mod} = 675$ μ V) under 20 Oe. **c**, Topographic image (I_t : 100 pA, V_b : 1.35 mV, and image size: 55×55 nm²) and simultaneously measured zero-bias dI/dV map ($V_{mod} = 135$ μ V) under 173 Oe.

In page 8 line 27, we add the discussion “Furthermore, we perform vortex measurement under magnetic fields, however, zero-bias-conductance (ZBC) maps do not show the existence of vortex, which may be due to the overlap of vortex [51] (see the details of analysis in Supplementary Note 18).”

We add the reference:

51. Tinkham, M. Introduction to superconductivity. Courier Corporation, (2004).

We add the discussion into SI as **Note 18**.

We add the references into SI as following:

8. Chen, D. Y. *et al.* Superconducting properties in a candidate topological nodal line semimetal SnTaS₂ with a centrosymmetric crystal structure. *Phys. Rev. B* **100**, 064516 (2019).
9. Brandt, E. H. Vortex-vortex interaction in thin superconducting films. *Phys. Rev. B* **79**,134526 (2009).
10. Abrikosov, A. A. An influence of the size on the critical field for type II superconductors. *Dokl. Akad. Nauk.* **86**, 489 (1952).
11. Zavaritskii, N. Superconducting properties of thallium and tin films condensed at low temperatures. *Dokl. Acad. Nauk.* **86**, 501 (1952).
12. Abrikosov, A. A. The magnetic properties of superconducting alloys. *J. Phys. Chem. Solids* **2**, 199 (1957).
13. Tinkham, M. Introduction to superconductivity. Courier Corporation, (2004).

Reviewer #3 (Remarks to the Author):

The manuscript submitted by Wenliang Zhu et al. reports a set of studies on the layered superconductor AuSn₄, including angular dependent transport, point-contact Andreev reflection spectroscopy, scanning tunneling microscopy/spectroscopy, angle-resolved photoemission spectroscopy, and first-principle calculation. The authors claim that their results support the presence of intrinsic surface *p*-wave superconductivity in the material. Although the authors have carried out extensive studies on the two-dimensional superconductor using various techniques and the results are certainly helpful in understanding of the superconductivity of the material, the manuscript contains fundamental flaws and the conclusions are highly debatable. Therefore, in my opinion, the present manuscript does not reach the standard of Nature Communications. My detailed comments are attached below.

Authors' reply: Thank this referee for his/her efforts and constructive suggestions on our manuscript, which are valuable and helpful for revising and improving our paper. According to these important suggestions, detailed replies have been made by carefully considering all comments.

Com. 1. The authors presented an angular-dependent magnetoresistance study of this type-II superconductor in various in-plane magnetic fields below the $H_{c2,\parallel}$ - this is very confusing because it is against the knowledge of type-II superconductivity. A type-II superconductor remains superconducting in a magnetic field below H_{c2} . So, this raises a question regarding how the authors define the H_{c2} at in this study a given temperature, but the definition is not explicitly explained in the manuscript. I assume that the authors take the onset of the resistivity drop as the field-temperature relation for the $H_{c2}(T)$, but this is not the conventional definition for a type-II superconductor. I believe, with a lot of respect, that the senior co-authors of the manuscript should have recognized this fundamental issue earlier.

Authors' reply: Thank the referee for this question, reminding us to give a definition

of the H_{c2} to avoid possible confusions. We agree with the referee that a type-II superconductor remains superconducting in a magnetic field below H_{c2} based on the conventional definition of H_{c2} and the onset of the resistivity drop as the field-temperature relation for the H_{c2} (T) is not the conventional definition for a type-II superconductor. In fact, for the convenience of discussing and describing experimental and physical issues, usually different definitions have been used in the literature. For example, under the given magnetic field H , the onset temperature, the zero-resistance temperature, the midpoint temperature of the resistive transition, *etc.*, are used to define H_{c2} [*Nature* **453**, 761–762 (2008), *Proc. Natl. Acad. Sci. USA.* **114**, 13144-13147 (2017), *Phys. Rev. Lett.* **116**, 127001 (2016)]. In this work, in order to study the symmetry of magnetoresistance in the range between the onset and the offset of superconductivity under a given magnetic field, we use the onset temperature of superconductivity to define H_{c2} .

In the revised manuscript, we have modified the inset of **Fig. 1d** to indicate the onset temperature (T_c) of superconductivity under a given magnetic field.

In page 4 line 29, we add “A sharp SC transition appears at the onset temperature $T_c \sim 2.4$ K, defined by the intersection of the tangent line and the R – T curves and indicated by the thin arrow (the inset of **Fig. 1d**).”

In page 5 line 27, we add “With standard four-probe method (**Fig. 2a**), we performed the angle-dependent magnetoresistance measurements by rotating magnetic field B in the ab -plane under the current I along a and b axis (see angular relationship in *Supplementary Fig. S17*).”

Com. 2. Continuing the comment above, the angular-dependent magnetoresistance measurements were actually performed in a field range where pairing fluctuation reduces the resistance (but not forming any superconducting path all the way through the material). The authors define the angle between the field direction and the crystal axis a as θ and show in **Fig. 2e** that the angular dependence of the

magnetoresistance when $I // b$ is 90 degrees rotated from that when $I // a$. But I don't understand why the case was made so complicated because if theta is simply defined as the angle between current and field (that is, regardless of the crystal lattice), the two angular dependences shown in Fig. 2e will be identical. Therefore, the "two-fold" angular dependence entirely depends on the relative angle between current and field. In my opinion, given the field range where the two-fold angle-dependent MR is observed, the in-plane anisotropy can be explained by the interplay of magnetic flux and pairing fluctuation. For instance, magnetic flux suppresses pairing fluctuation, but the strength/level of the suppression changes with the angle between the field and the current. In fact, it would be interesting to see that at a given in-plane field, whether the zero-resistance temperature is related to the angle between current and field.

Authors' reply: Thank the referee for this valuable comment, reminding us to give a clearer description of the experiment and correct our careless mistakes to understand two angular dependences. Besides, according to the referee's suggestions, we give the angle-dependent zero-resistance temperature at a given in-plane field, as what follows:

1) It is well known that the angle-dependent magnetoresistance in the range between the onset and the offset of superconductivity can be used to understand the symmetry of the superconducting state [*Nat. Phys.* **17**, 949–954 (2021), *Nat. Commun.* **14**, 3046 (2023), *Nat. Phys.* **19**, 106–113 (2023)]. However, there are two key conditions that need to be fulfilled before using this method. These two key conditions are that the normal state of superconductor has high symmetry (preferably isotropy) and low MR = $[\rho_{xx}(B) - \rho_{xx}(0)]/\rho_{xx}(0) \times 100\%$. Besides, the most important one is that the symmetry of magnetoresistance in the range between the onset and the offset of superconductivity should be related to the angle between the crystal axis and field, not the angle between current and field.

2) In this work, we found that the normal state of AuSn₄ show almost isotropic magnetoresistance (**Figs. 2b 2c and 2d** in revised manuscript) and extremely small MR only ~0.02% (extracted from **Fig. 1f** and **Fig. 2d** in revised manuscript) at 600 Oe,

which indicates the angle-dependent magnetoresistance of AuSn₄ in the range between the onset and the offset of superconductivity may be suitable to study the symmetry of the superconducting state. Therefore, in order to exclude extrinsic factors, we performed the angle-dependent magnetoresistance measurements with the current I along a and b axis, as shown in **Fig. R8a** and **8b**, respectively. Here, θ is the angle between the directions of fields B and current I , not the angle between the directions of fields B and a axis that we made the careless mistakes in manuscript. When $\theta = 0^\circ$, it is worth to note that the current I is parallel to the b axis in **Fig. R8a** and the current I is parallel to the a axis in **Fig. R8b**. The fact that the two angular dependences are not identical, giving the reason why we persist that the observed symmetry originates from the superconducting state.

Fig. R8. Schematic measurement configuration under in-plane magnetic fields. **a** and **b**, the current I is parallel to b and a axis, respectively. θ is defined as the angle between the fields B and current I .

3) According to the suggestion, we give the temperature-dependent resistivity under a given in-plane field 120 Oe with the current I along a and b axis, as shown in **Figs. R9 a-c** and **Figs. R9 d-e**, respectively. Here, the zero-resistance temperature (T_c^0) is defined as the intersection of the tangent line and the R - T curve, indicated by the thin arrow. **Figure R9f** shows the angle dependence of the zero-resistance temperature extracted from **Figs. R9b-e**. We find that the two angular dependences in **Fig. R9f** are not identical, consistent with the results of the angle-dependent magnetoresistance in the range between the onset and the offset of superconductivity.

In page 5 line 27, we add "With standard four-probe method (**Fig. 2a**), we performed the angle-dependent magnetoresistance measurements by rotating magnetic field B

in the ab -plane under the current I along a and b axis (see angular relationship in **Supplementary Fig. S17**)." for detailed descriptions of the angle-dependent magnetoresistance measurements.

In page 6 line 15, we add "The zero-resistance temperature also shows two-fold modulation and an 90° angle between the C_2 axes for $I \parallel a$ and $I \parallel b$ (see **Supplementary Fig. S18**).".

In the caption of **Fig.2** we correct our careless mistakes. Meanwhile, **Figs. R8-9** were added in the revised Supplementary as **Note 15** and **Figs. S17-18**.

Fig. R9. | Two-fold angular dependence of the zero-resistance temperature of superconducting AuSn₄ under a given in-plane magnetic fields. a-c, The temperature-dependent resistivity with the current I along b axis under 120 Oe and selected angel. d-e, The temperature-dependent resistivity with the current I along a axis under 120 Oe and selected angel. f, The angle-dependent T_c^0 extracted from the panels of b-e.

Com. 3. There is a long history of debate on whether a zero-bias conductance peak (ZBCP) can be properly related to unconventional superconductivity in point-contact Andreev reflection measurements. For example, Gifford et al. [J. Appl. Phys. 120, 163901 (2016)] has a detailed study of ZBCP observed in point-contact spectroscopy measurements. The present manuscript suggests that a two-fold angular dependence of the ZBCP at a field of 120 Oe where AuSn₄ remains superconducting is a signature of unconventional Andreev bound state. However, similar to Comment 2 described above, the in-gap features may change due to the variation of the interaction between the penetrated magnetic flux (assuming the external magnetic field is above H_{c1}) and the paired electrons. Given the device design shown in the inset of Fig. S4, this possibility cannot be ruled out in the present study. Also, it is very unusual to relate a pair of dips (± 3 mV), instead of peaks, to superconductivity or Andreev bound states in point-contact measurements. In addition, I also notice that the contact conductance increased by only $\sim 4\%$ at zero bias. Although one may find some reports having similar “enhancement” in literature, this is not an ideal Andreev-reflection junction. In such a case, many artificial/accidental features may be exaggerated, and analysis should not rely on such small features.

Authors' reply: Thank the referee for these comments, which are very valuable and helpful for revising and improving our manuscript. According to these comments, we have made corresponding revisions in the revised manuscript. And detailed explanations are presented as the following three points.

1) As the referee said, that there is a long history of debate on whether a zero-bias conductance peak (ZBCP) can be properly related to unconventional superconductivity in

point-contact Andreev reflection measurements, and likely that the outstanding work [*J. Appl. Phys.* **120**, 163901-9 (2016)] claimed that the point contact spectrum can be influenced by the size of contact area between tip and sample. Thus, we performed a lot of point-contact measurements and tried to minimize the size of contact area between Ag paste and sample. Usually, we can obtain point contact spectroscopy in the ballistic regime with this method. Despite our extensive efforts, we only observed ZBCP with the contact conductance increased by $\sim 4\%$ at zero bias, which is very interesting to us. With considering the reference [*J. Appl. Phys.* **120**, 163901 9 (2016)], the ZBCP maybe originate from the property of depaired electrons instead of Andreev bound state. Hence, if the ZBCP is generated in this way, the symmetry should be coincident with the normal state. However, we found that the ZBCP at 0 mV shows a two-fold symmetry with rotating the in-plane magnetic field, consistent with the angle-dependent magnetoresistance in the range between the onset and the offset of superconductivity and the angle dependence of the zero-resistance temperature, but not with the angle-dependent magnetoresistance in the normal state and the ZBCP at 6 mV. Based on these evidences, we persisted that the ZBCP in our experiment cannot be attributed to the depairing effect mentioned in the reference [*J. Appl. Phys.* **120**, 163901 9 (2016)] and its two-fold symmetry should originate from superconducting state.

Here, although these results indicated that the superconducting state dominates the ZBCP, other factors also should be considered according to this comment. Therefore, we made corresponding revision in the revised manuscript.

Two thin arrows marked in **Fig. 2h** are just used to indicate the minimum value of dips. These two dips were not related to the superconductivity or Andreev bound states in point-contact measurements. In order to avoid possible confusion, we have removed these two arrows from **Fig. 2h** and made corresponding revisions in the revised manuscript.

2) As for the near $\sim 4\%$ conductance increasing, this is not puzzling and on the contrary that may be evidence for surface *p*-wave superconductivity. Firstly, although a lot of point-contact measurements were performed, but we cannot obtain more than about 4% conductance increasing, indicating that it may be intrinsic. Secondly, similar observation

also appears in $\text{Cu}_x\text{Bi}_2\text{Se}_3$ system [*Phys. Rev. Lett.* **107**, 217001 (2011)], which confirms the topological superconductivity. The superconductivity in AuSn_4 is very similar to that in $\text{Cu}_x\text{Bi}_2\text{Se}_3$, in both the bulk superconductivity is full-gap but the surface superconductivity is gapless. In the point-contact experiment, both the bulk and surface superconductivity can be detected. The mixing between bulk full-gap *s*-wave superconductivity and surface gapless *p*-wave superconductivity causes the ZBCP and two-fold symmetry. Otherwise, if only one component can be detected, for example, for a *p*-wave component, the point-contact spectroscopy should display a very sharp peak and also a two-fold symmetry for zero-energy conductance; for a *s*-wave component, the spectrum should be full-gap and four-fold symmetry; However, neither of these have not been observed in our experiment. Excitingly, we find that the angle dependence of point-contact spectroscopy at 0 mV is consistent with the angle-dependent magnetoresistance in the range between the onset and the offset of superconductivity and the angle dependence of the zero-resistance temperature, but not with the angle-dependent magnetoresistance in the normal state and the ZBCP at 6 mV. So the near $\sim 4\%$ conductance increasing reflects the mixing of *s*-wave and *p*-wave component and such small feature is intrinsic.

In order to make the results of the point-contact experiments clearer, the descriptions and discussions of **Fig. 2g** and **2i** have been moved from the page 7 line 17 to the page 6 line 26.

Meanwhile, at the page 6 and 7 of the revised manuscript, we also have made detailed revisions to better understand the results of the point-contact experiments (marked in red) and added the reference "Gifford, J. A. *et al.* Zero bias anomaly in Andreev reflection spectroscopy. *J. Appl. Phys.* **120**, 163901 (2016)." as [44].

Com. 4. The STS data shown in Fig. 3 indicate a gap-like feature (called "outer gap" in the text) at ~ 3.2 mV in addition to the smaller gap feature at ~ 1.5 mV, and the authors claim it as evidence of multi bands, presumably referring to multi-band superconductivity. However, the "outer gap" is very weak and cannot be distinguished

from the “pseudogap-like” feature that survives far above the T_c or H_c of the material. Therefore, I feel that the conclusion of multiband lacks sufficient support in the STS measurements. In addition, the zero-bias conductance of the tunneling spectrum is only about 8% smaller than the normal conductance, and in such a case, scattering-induced spectrum broadening can introduce a significantly large error for the fitting parameters. I notice that although the fitting curves in Fig. 3c and Fig. S11 d have similar “shape”, the magnitudes of the dips are completely different. The authors should list all the fitting parameters in the manuscript.

Authors' reply: We thank the referee for this constructive comment and quite agree with this referee. As we reply the comment 1 of Referee #1, we think that the large gap size and dual-gap structure, may be originated from the thermal broadening and large lock-in modulation bias. In order to eliminate the thermal broadening effect, we carried out the dI/dV measurements at a lower temperature ~ 160 mK, by collaborated with Prof. Shaochun Li in Nanjing University, China. In **Fig. R1**, a single SC gap is clearly observed on the flat terrace of AuSn_4 surface at 160 mK. The gap size is about 0.487 meV, with the depth of gap about 87%. Note, we adopt much smaller modulation bias as $V_{mod} = 40 \mu\text{V}$.

Fig. R1. A representative dI/dV spectra taken at 160 mK and 0 Oe. The dI/dV curve measured on most of terraces. ($V_b = 1.35$ mV, $I_t = 200$ pA, $V_{mod} = 40 \mu\text{V}$).

In SI, we remove the section of the SC gap fitting.

In main content, we update **Fig. 3c** with our new STS results.

In page 8 line 11, we revise to “A typical normalized dI/dV spectrum measured at 0.16 K (**Fig. 3c**) shows a superconductive gap feature, a deep V-shaped gap about $\Delta \sim 0.487$ meV (defined as half of the gap width) with the depth of gap about 87%.”

In page 8 line 20, we revise to “The ratio $2\Delta(0)/k_B T_c$ (k_B is the Boltzmann constant) is estimated to be ~ 4.7 , which is in the range of the medium coupling Bardeen–Cooper–Schrieffer (BCS) superconductors. ”

In the caption of Fig. 3, we rewrite “A representative dI/dV spectrum taken at 0.16 K, showing a superconducting gap with the size and depth of gap about 0.487 meV and 87%, respectively. Here, we adopt small lock-in modulation bias (V_{mod}) ~ 40 μ V in order to eliminate the broadening effect.” And add “Note, the STS taken at 0.38 K with 0.3 mV V_{mod} , give a dual-gap feature with enlarged gap size and shallower depth, probably due to the thermal broadening effect and large lock-in modulation bias.”

Reviewers' Comments:

Reviewer #1:

Remarks to the Author:

I have read the authors' reply and the revised manuscript. The authors have performed new STM measurement at much lower temperature, which revealed a clear superconducting gap in STS and clarified their previous results. The gap size and depth are now reasonable. They also provide more details on the stripe-like pattern and its modulation effect on SC gap. The results now more clearly suggest there could be unconventional superconductivity induced by a mixing of $s + p$ pairing in the sample. Therefore I would recommend its publication on Nature communications.

Reviewer #2:

Remarks to the Author:

I think that the authors have reasonably answered my questions regarding (i) the relationship between the 1D-type CDW-like modulation and band structure, (ii) the influence of superstructure to the superconducting gap, (iii) the agreement of ARPES data with the DFT calculation for monolayer, and (iv) the k_z dependence of the band structure.

Regarding the Majorana mode at the edge shown in Fig. R7, the dI/dV spectrum exhibits much smaller change than I have expected. It would be useful if the authors can show that the difference in the shape of density of states (DOS) between edge and terrace is much larger than the spatial modulation of the DOS on the single terrace.

It is a pity that authors could not resolve clearly the vortex core. But I understand experimental difficulty on this point.

In my opinion, the manuscript is appropriately revised by reflecting comments from the reviewers. Now I recommend the publication of this manuscript in Nature Communications with minor revision.

Reviewer #3:

Remarks to the Author:

After reading the authors' reply letter and updated manuscript, I believe that the manuscript has been greatly improved and my previous concerns have been properly addressed in these files. I would recommend this updated manuscript to be published in Nature Communications.

Dear Referees:

We thank the referees for their thoughtful review of our manuscript entitled “An Intrinsic Surface p -wave Superconductivity in Layered AuSn₄”. We are glad to see the positive assessment from the reviewers. We benefit a lot from these suggestions and comments, and we revised the manuscript accordingly. The detailed point-to-point responses are provided as follows, marked in blue. The modifications in the main text are marked in red.

With all the concerns have been addressed here and in the revised manuscript, we hope that the manuscript is now appropriate to be published in *Nature Communications*.

Sincerely,

Minghu Pan, Shao-Chun Li, Ping Zhang, Genfu Chen and Qi-Kun Xue

Manuscript NCOMMS-23-24487-T

REVIEWER COMMENTS

Reviewer #1 (Remarks to the Author):

I have read the authors' reply and the revised manuscript. The authors have performed new STM measurement at much lower temperature, which revealed a clear superconducting gap in STS and clarified their previous results. The gap size and depth are now reasonable. They also provide more details on the stripe-like pattern and its modulation effect on SC gap. The results now more clearly suggest there could be unconventional superconductivity induced by a mixing of $s + p$ pairing in the sample. Therefore I would recommend its publication on Nature communications.

Authors' reply: Thank the referee for his/her efforts and recommendation for publication of this manuscript.

Reviewer #2 (Remarks to the Author):

I think that the authors have reasonably answered my questions regarding (i) the relationship between the 1D-type CDW-like modulation and band structure, (ii) the influence of superstructure to the superconducting gap, (iii) the agreement of ARPES data with the DFT calculation for monolayer, and (iv) the k_z dependence of the band structure.

It is a pity that authors could not resolve clearly the vortex core. But I understand experimental difficulty on this point.

In my opinion, the manuscript is appropriately revised by reflecting comments from the reviewers. Now I recommend the publication of this manuscript in Nature Communications with minor revision.

Authors' reply: We thank the referee for his/her valuable suggestion and recommendation for publication of this manuscript.

Com. Regarding the Majorana mode at the edge shown in Fig. R7, the dI/dV spectrum exhibits much smaller change than I have expected. It would be useful if the authors can show that the difference in the shape of density of states (DOS) between edge and terrace is much larger than the spatial modulation of the DOS on the single terrace.

Authors' reply: According to this comment, we compare the shapes of DOS between the edge and terrace, the depth and size of SC gap at the edge are suppressed to 82% and 0.446 meV, compared to the values on the terrace (87.8% and 0.516 meV). The variation of SC gap due to CDW modulation on the terrace are much small, 0.486-0.516 meV of gap size without observable change in the depth.

In page 8 line 6, we added "The depth and size of SC gap at the edge are both suppressed to 82% and 0.446 meV, compared to the values on the terrace (87.8% and 0.516 meV). The variation of SC gap due to CDW modulation on the terrace are much smaller, 0.486-0.516 meV of gap size without observable change in the depth."

Fig. R1. The correlations between SC gap and CDW modulations. **a**, Topographic image (I_t : 100 pA, V_b : 1.0 V, and image size: $20 \times 20 \text{ nm}^2$) taken at temperature of 4.2K. **b-c**, the line dI/dV spectroscopic survey taken along the blue (black) arrows in panel **a**. Inset: intensity map around the coherence peak energy, clearly showing the modulation of the coherence peaks in energy. The STS set-up conditions: **b**: $V_b = 2.7 \text{ mV}$, $I_t = 200 \text{ pA}$, $V_{mod} = 40 \text{ } \mu\text{V}$, **c**: $V_b = 1.35 \text{ mV}$, $I_t = 200 \text{ pA}$, $V_{mod} = 27 \text{ } \mu\text{V}$ taken at the temperature of 160 mK. **d**, The comparison of the variation of dI/dV spectra due to the spatial modulation.

Fig. R2. STS measured on the terrace and at the edge. **a-b**, Topographic images showing the locations of point dI/dV spectra (**a**: I_t : 100 pA, V_b : 100 mV, and image size: $130 \times 130 \text{ nm}^2$, **b**: I_t : 100 pA, V_b : 0.675 V, and image size: $130 \times 130 \text{ nm}^2$). **c**, the comparison of dI/dV spectra on the terrace (red) and at the edge (blue). The STS set-up conditions: edge: $V_b = 6.75 \text{ mV}$, $I_t = 100 \text{ pA}$, $V_{mod} = 67.5 \text{ } \mu\text{V}$, terrace: $V_b = 1.35 \text{ mV}$, $I_t = 300 \text{ pA}$, $V_{mod} = 27 \text{ } \mu\text{V}$, taken at the temperature of 160 mK and magnetic field of 20 Oe.

Reviewer #3 (Remarks to the Author):

The After reading the authors' reply letter and updated manuscript, I believe that the manuscript has been greatly improved and my previous concerns have been properly addressed in these files. I would recommend this updated manuscript to be published in Nature Communications.

Authors' reply: Thank the referee for his/her efforts and recommendation for publication of this manuscript.